# FAT-switch-based quantitative S-nitrosoproteomics reveals a key role of GSNOR1 in regulating ER functions

Guochen Qin [1,2], Menghuan Qu[1,3], Bei Jia[1], Wei Wang[4], Zhuojun Luo[5], Chun-Peng Song [4], W. Andy Tao [5,6] & Pengcheng Wang [7] ✉

Reversible protein S-nitrosylation regulates a wide range of biological functions and physiological activities in plants. However, it is challenging to quantitively determine the S-nitrosylation targets and dynamics in vivo. In this study, we develop a highly sensitive and efficient fluorous affinity tag-switch (FAT-switch) chemical proteomics approach for S-nitrosylation peptide enrichment and detection. We quantitatively compare the global S-nitrosylation profiles in wild-type Arabidopsis and *gsnor1/hot5/par2* mutant using this approach, and identify 2,121 S-nitrosylation peptides in 1,595 protein groups, including many previously unrevealed S-nitrosylated proteins. These are 408 S-nitrosylated sites in 360 protein groups showing an accumulation in *hot5-4* mutant when compared to wild type. Biochemical and genetic validation reveal that S-nitrosylation at Cys337 in ER OXIDOREDUCTASE 1 (ERO1) causes the rearrangement of disulfide, resulting in enhanced ERO1 activity. This study offers a powerful and applicable tool for S-nitrosylation research, which provides valuable resources for studies on S-nitrosylation-regulated ER functions in plants.

Protein S-nitrosylation is a reversible post-translational modification, which plays a major regulatory role in NO-related and redox signaling pathways through the covalent interaction of nitric oxide (NO) with the thiol group of cysteine residues[1,2]. Extensive studies have shown that S-nitrosylation regulates multiple physiological processes including abiotic stress responses, phytohormone signaling, immune responses, plant growth and development[3–7], through modulating stability, activity, subcellular localization, conformation change, or protein-protein interaction of target proteins[1,2,8,9].

S-nitrosoglutathione (GSNO) is a primary endogenous NO donor. In Arabidopsis, a single copy of GSNO reductase 1 (GSNOR1) dominantly controls the endogenous GSNO level and the global protein S-nitrosylation level[10]. The loss-of-function mutants of GSNOR1, *gsnor1/sensitive to hot temperatures5* (*hot5*)/ *paraquat resistant2* (*par2*), show severe developmental abnormalities in both reproductive and vegetative stages[7,11,12], and altered responses to phytohormones such as auxin[13], abscisic acid (ABA)[14], and strigolactones[15]. Accordingly, the *gsnor1/hot5/par2* mutants are ideal subjects for the studies on S-nitrosylation in plants since they show altered responses to environmental changes such as heat, drought, hypoxia, and pathogens[6,11,14,16]. Previous studies have identified several S-nitrosylation targets with S-nitrosylation accumulated in *gsnor1* mutants and revealed the mechanisms by which S-nitrosylation regulates development, immunity, and stress responses in plants. For

[1]Shanghai Center for Plant Stress Biology, CAS Center for Excellence in Molecular Plant Sciences, Chinese Academy of Sciences, 200032 Shanghai, China. [2]Peking University Institute of Advanced Agricultural Sciences, Shandong Laboratory of Advanced Agricultural Sciences at Weifang, 261000 Weifang, Shandong, China. [3]University of Chinese Academy of Sciences, Beijing, China. [4]State Key Laboratory of Crop Stress Adaptation and Improvement, School of Life Sciences, Henan University, 475004 Kaifeng, China. [5]Department of Biochemistry, Purdue University, West Lafayette, IN 47907, USA. [6]Department of Chemistry, Purdue University, West Lafayette, IN 47907, USA. [7]Institute of Advanced Biotechnology and School of Life Sciences, Southern University of Science and Technology, 518055 Shenzhen, China. ✉e-mail: wangpc@sustech.edu.cn

example, the S-nitrosylated salicylic acid receptor NPR1, NADPH oxidase RBOHD, APX1, and SGR1 were involved in plant immunity to various pathogens[4,5,17,18]. In addition, the S-nitrosylated bacterial effector HopAI1 inhibits its phosphothreonine lyase activity and HR responses[19]. Furthermore, the NO negatively regulates gibberellin signaling via S-nitrosylation of RGA, a key member of DELLA proteins, to coordinate with the balance of growth and stress responses[20]. The GSNOR1 protein is also S-nitrosylated at Cys10 residue, and the S-nitrosylated Cys10 residue functions as a feedback controller in GSNOR1 activity in hypoxia responses[16]. However, a limited number of S-nitrosylation targets were identified specifically, and limited S-nitrosylated proteins enriched in *gsnor1* mutant were recognized. A comprehensive analysis of S-nitrosylation in *gsnor1* mutant will expand our knowledge and provide insight into the broad role of protein S-nitrosylation in plants.

So far, it remains challenging to quantitively determine the global S-nitrosylation, especially in plants, partly due to the labile nature of the S-NO bond and the relatively low abundance of endogenously S-nitrosylated proteins. Currently, a biotin-switch method combined with mass spectrometry (MS) is the major strategy for identifying protein S-nitrosylation[21]. By this method, free cysteines are first blocked by a thiol-reactive reagent, such as methyl methanethiosulfonate (MMTS) or iodoacetamide (IAM). The S-NO bounds are specifically reduced in the presence of ascorbate and chemically substituted by the biotin-containing affinity molecule, such as biotin-HPDP, PEO-iodoacetyl-biotin or biotin-maleimide[22]. After further purification by biotin-affinity isolation, the biotinylated proteins are identified by MS. The proteome-wide identification of S-nitrosylation by this method has been reported in the model plant Arabidopsis[23]. However, biotin related reagents are difficult to elute from the capture resin, which can complicate tandem mass spectrometry spectral interpretation[24]. As a result, only several hundred S-nitrosylated proteins could be identified by the biotin-switch or iodo-TMT methods in plants[23] (see the summary Supplementary Table 1), limiting the deeper profiling of S-nitrosylation. Several other labeling reagents, such as isotope coded affinity tag (ICAT)[25], iodo-TMT[26,27], cysteine-specific phosphonate adaptable tag (CysPAT)[28], bioorthogonal cleavable-linker (Cys-BOOST)[29], and fluorous affinity tag (FAT)[30], were also used to label and enrich S-nitrosopeptides. However, these commonly practiced methods were rarely applied to plants.

Among these reagents, FAT with covalent C-F bonds avoids unexpected dissociation during the MS process and reduces the complexity of tandem MS, allowing the identification of low abundance targets[30]. FAT has been successfully applied in some proteomic studies of post-translational modification, such as protein phosphorylation[31], tyrosine nitration[32], HNE modification[33], and S-nitrosylation[30]. However, whether FATs are applicable in plants to acquire in-depth S-nitrosylation proteomic analysis requires further testing.

In this study, we developed an N-(4,4,5,5,6,6,7,7,8,8,9,9,9-tridecafluorononyl) iodoacetamide (TFIA) based FAT-switch approach, which shows a significantly improved sensitivity and efficiency of enrichment and detection of S-nitrosylated peptides than the classic biotin-switch method. We quantitatively profiled the S-nitrosoproteome in wild type and *gsnor1/hot5/par2* mutant seedlings through combining the TFIA based FAT-switch approach with the tandem mass tag (TMT)-labeling strategy. Among a total of 1,559 quantitative nitrosopeptides, 408 nitrosopeptides, representing 360 proteins, were significantly upregulated in the *gsnor1* mutant than in the wild type. These S-nitrosylated proteins accumulated in *gsnor1* mutant are mainly involved in metabolism, stress response, cellular component organization, signal transduction, etc. Based on the nitrosoproteomics, we revealed that GSNOR1 participates in endoplasmic reticulum (ER) stress and unfolded protein response (UPR) by increasing the S-nitrosylation of ER OXIDOREDUCTASE 1 (ERO1) and other proteins. The S-nitrosylation at Cys337 in ERO1 leads to the rearrangement of disulfide groups and enhances ERO1 activity. These results provide a comprehensive view of the role of S-nitrosylation in controlling cellular activities and offer important resources for further studies on the mechanisms underlying plant metabolism and responses to environmental stresses.

## Results

### Development of FAT-switch method for S-nitrosoproteome

In this study, we developed a FAT-switch method, which combined the FAT labeling S-nitrosylated proteins and the fluorous solid-phase extraction of target peptides. The overall scheme is shown in Fig. 1a. First, the S-alkylating reagent IAM was used to block free cysteine thiols allowing the S-nitrosylated cysteine and disulfide bonds to keep intact. After the removal of excess IAM by acetone precipitation, the S-nitrosylated cysteines were reduced explicitly by a low concentration of ascorbate (Asc) and substituted by a sulfhydryl reactive fluorous reagent, N-(4,4,5,5,6,6,7,7,8,8,9,9,9-tridecafluorononyl) iodoacetamide (TFIA). The IAM-blocked FAT-labeled proteins were subjected to trypsin digestion and further enriched by the C18 column at a low acetonitrile (ACN) concentration. After further enrichment by nanographite fluoride (nGF), the FAT-labeled peptides were subjected to liquid chromatography-tandem mass spectrometry (LC-MS/MS) detection. We first used the recombinant PYRABACTIN RESISTANCE1-LIKE 4(PYL4), a well-defined S-nitrosylated protein, to measure the efficiency of labeling of this workflow in vitro. One of the three free cysteines in PYL4, Cys185, is the known S-nitrosylated site[34]. After GSNO treatment and free cysteine blocking, we used a commercial TFIA as the fluorinated reagent to label S-nitrosylated sites (Supplementary Fig. S1a), and successfully detected a characteristic mass shift (417.04 Da) of labeled Cys185 by LC-MS/MS.

We then performed the labeling reaction under different concentrations of TFIA and different incubation time using total protein extract from 10-day-old Arabidopsis seedlings. The results showed that the highest efficiency of S-nitrosylated peptide enrichment was obtained with 0.2 μg/μL of TFIA with 2 h of labeling, which allowed us to detect 1,920 TFIA-labeled peptides in a single run of LC-MS/MS assay (Supplementary Fig. 1b, c). We also discovered that adding a step of C18 enrichment significantly reduced the non-specific peptides and increased the purified percentage to about 30% (Supplementary Fig. 1d, e). Together, these results suggested that the FAT-switch method is a highly sensitive and efficient method for the enrichment and detection of S-nitrosylated peptides in plants.

We further compared the sensitivity and efficiency of FAT-switch and classical biotin-switch methods of the enrichment and detection of S-nitrosylation peptides using the same batch of plant tissues. Compared to 771 S-nitrosylated peptides detected by the biotin-switch method, the FAT-switch method detected 2,559 S-nitrosylated peptides from 0.2 g of seedlings from three biological replicates (Fig. 1b; Supplementary Data 1 and 2). As shown in Fig. 1c, the average MS/MS intensity observed in the FAT-switch method was approximately 10-fold higher than the biotin-switch assay result, suggesting that the FAT-switch method showed a 7.4-fold higher sensitivity (3.54 vs. 0.48, $p < 0.001$, two-tailed paired $t$ test) compared to the biotin-switch method in the context of identified S-nitrosylated peptide per microgram of tissues (Fig. 1d). We also compared the specificity of biotin-switch and FAT-switch methods on S-nitrosylated protein by parallel assays with or without Asc. As shown in Supplementary Fig. 1f, g, the results showed that the number and intensity of S-nitrosylated peptides were significantly elevated in the presence of Asc (Supplementary Fig. 1g). Taken together, these results suggest that our FAT-switch method has achieved a broad-scaled S-nitrosylation proteomic analysis with significantly enhanced sensitivity, efficiency and comparable specificity of enrichment and identification of global S-nitrosylated peptides in plants.

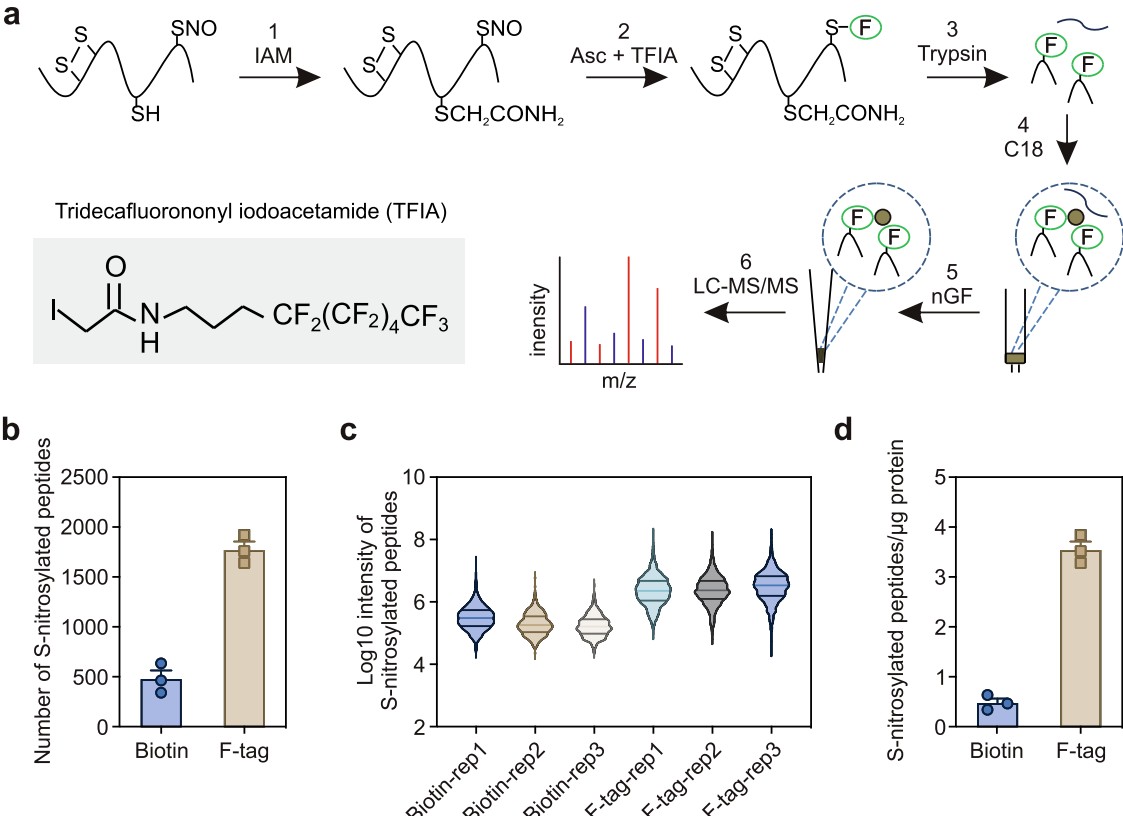

**Fig. 1 | Development of FAT-switch approach for the detection of S-nitrosoproteome in *Arabidopsis*. a** Reaction schema for FAT-switch approach to label S-nitrosylated proteins and enrich target peptides. Step 1, free cysteine thiols were blocked by IAM. In step 2, the S-nitrosylated cysteines were specifically reduced by a low concentration of ascorbate and substituted by a sulfhydryl reactive fluorous tag reagent. Step 3, proteins were digested by trypsin. After then, C18 column were used to capture fluorous-tag labeling peptides and remove high hydrophilic peptide by a low ACN concentration of washing buffer (Step 4). Finally, fluorous labeled peptides were further purified by nano-graphite fluoride, and identified by LC-MS/MS (Step 5 and 6). **b** The number of S-nitrosylation peptides identified by biotin switch method and FAT-switch approach with same bench of triplicate samples. Error bars, SD ($n = 3$ biological replicates). **c** Boxplots of $\log_{10}$ MS/MS intensity of all identified S-nitrosylation peptides in biotinswitch method and FAT-switch approach with same bench of triplicate samples. **d** Number of S-nitrosylation peptides identified per µg of Arabidopsis proteins. Error bars, SD ($n = 3$ biological replicates).

## Quantitative comparison of S-nitrosoproteome in wild type and *gsnor1*

We then coupled the TMT-labeling strategy and FAT-switch procedure to quantitatively profile the S-nitrosoproteome in Ws wild type and *hot5-4* mutant. After FAT-switch of S-nitrosylated protein and enzymatic digestion, triplicates of wild type and *hot5-4* samples were labeled with TMT reagent and mixed (Supplementary Fig. 2a). S-nitrosylated peptides were then enriched with the C18 column and home-made nGF column, followed by 4-hour LC-MS/MS analysis. A total of 2,121 S-nitrosylation peptides of 1,595 protein groups from either wild type or the *hot5-4* mutant seedlings were identified, and 1,559 unique S-nitrosylated peptides of 1,231 protein groups were quantifiable (Supplementary Data 3).

By comparison, 41 of 105 (39.0%) putative S-nitrosylated proteins identified in Lindermayr et al.[35], 75.5% (40 out of 53) S-nitrosylated peptides in Fare et al.[36], 56.5% (35 out of 62) S-nitrosylated peptides in Puyaubert et al.[37], and 46.3% (20 out of 41) peptides in Lawrence II et al.[38] were included in the list of S-nitrosylation peptides identified by the FAT-method in our study (Supplementary Data 4). 247 out of 1,189 (20.6%) S-nitrosylated peptides identified by Hu et al.[23] were presented in the list of S-nitrosylation peptides in our study (Supplementary Fig. 3a). A possible explanation for the differential S-nitrosylated peptides identified in Hu et al.[23] and in this study could be that the FAT-TMT method enriched more hydrophobic peptides than the Biotin-switch method (Supplementary Fig. 3b). Previously known S-nitrosylated proteins including APX1, clCDH, PrxII E, and SABP3 are

present in our S-nitrosproteomics results (Supplementary Data 4). These results demonstrate that our study has covered the majority of S-nitrosylated peptides identified by either large-scale proteomics or low-throughput biochemistry studies, further suggesting the high sensitivity and reliability of our FAT-switch-method for S-nitrosoproteomics studies in plants. Missing of some known S-nitrosylated targets might be because of extreme low abundance in our tested samples or their peptides are inappropriate for LC-MS/MS detection.

We then performed a pair-wise comparison of S-nitrosylated-peptides in wild type and *hot5-4* mutant. The 408 S-nitrosylated peptides of 360 unique proteins were up-regulated in the *hot5-4* mutant compared to the wild type (fold-change > 1.5, $p < 0.05$, two-tailed unpaired $t$ test) (Fig. 2a; Supplementary Data 5). To validate the quantitative S-nitrosoproteomics results, we randomly selected several proteins which have commercial antibodies. We measured the S-nitrosylation levels of the selected protein candidates in Ws and *hot5-4* mutant seedlings by biotin-switch of S-nitrosylated proteins enrichment and subsequential immunoblotting. All of the five tested proteins, RPN12a, PSB33, SDIRIP1, CRT, and CNX1 showed higher S-nitrosylation level in *hot5-4* than in the Ws wild type (Fig. 2b and below). These results have shown the reliability of the quantitative S-nitrosoproteomic analyses comparing in wild type and the *hot5-4* mutant.

Interestingly, we observed that the abundances of 71 S-nitrosylated peptides (fold-change <0.7, $p < 0.05$, two-tailed

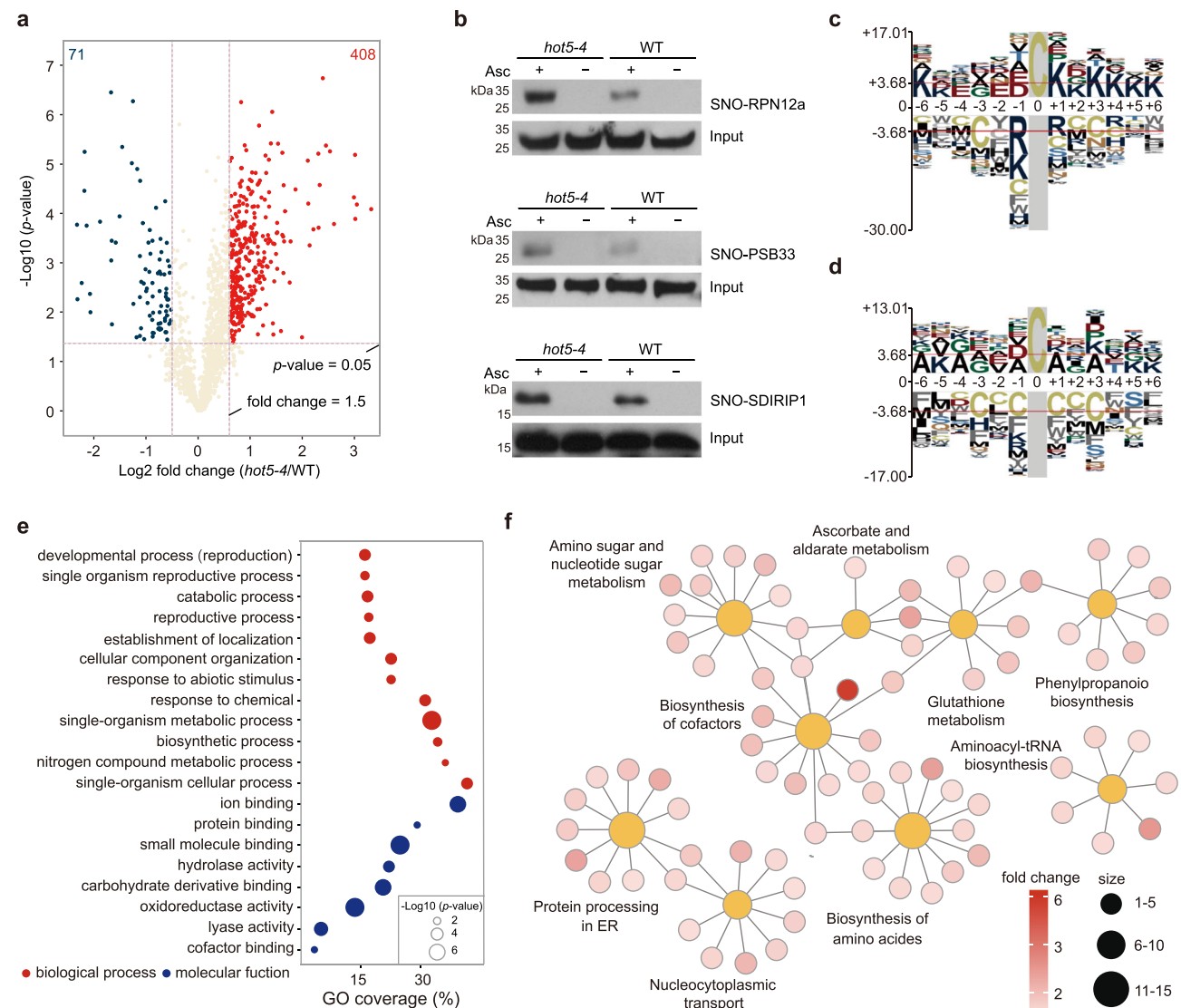

**Fig. 2 | Quantitative analysis of the S-nitrosoproteome in wild type and *hot5-4*.** **a** Volcano plot of S-nitrosylation sites identify in wild type and *hot5-4* mutant. Red dots (*n* = 408) represent S-nitrosylated peptides enriched in *hot5-4*, (fold change > 1.5, *p* < 0.05, two-tailed unpaired *t* test). Blue dots (*n* = 71) represent S-nitrosylated peptides enriched in the wild type (fold change < 0.67, *p* < 0.05, two-tailed unpaired *t* test). **b** Analysis of S-nitrosylation of RPN12a, PSB33 and SDIRIP1 proteins in wild type and *hot5-4* seedlings by in vivo biotin-switch method. The blots were detected by the anti-RPN12a, anti-PSB33 and anti-SDIRIP1 antibodies, respectively. Images shown are representative of at least three independent experiments. **c** Motif analysis of 408 S-nitrosylation sites enriched in *hot5-4* mutant. Overrepresentation motifs of Cys-SNO sequence windows (−6 to +6 residues) constructed by pLogo. **d** S-nitrosylation motif analysis of 379 none up-regulated S-nitrosylation sites (0.75 <fold change <1.2) in *hot5-4* mutant. Over-representation motifs of Cys-SNO sequence windows (−6 to +6 residues) constructed by pLogo. **e** Biological process and molecular function distribution of 360 up-regulated S-nitrosylated proteins (*p*-value, Fisher's exact test). **f** Enriched KEGG pathway in *hot5-4* accumulated S-nitrosylated proteins.

unpaired *t* test), representing 62 unique proteins, were lower in *hot5-4* mutant than in the wild-type seedlings (Supplementary Data 6). We performed a label-free quantitative proteomic analysis to investigate the global protein abundances in wild type and *hot5-4* mutant. As the result, 36 down-regulated S-nitrosylated proteins were quantified in three biological replicates of proteomics analyses. Among these, 31 proteins were significantly downregulated (fold-change <0.7, *p* < 0.05, two-tailed unpaired *t* test) in *hot5-4* than in wild type (Supplementary Data 6). These results suggest that the reduced protein level is the main reason for the downregulated S-nitrosylated sites in *hot5-4* mutant.

To investigate the properties of S-nitrosylation sites, we compared the position-specific frequencies of the amino acid residues surrounding S-nitrosylated Cys residues of 408 upregulated peptides and 379 unchanged peptides (fold-change from 0.7 to 1.2) by

the pLogo motif visualizing program[39]. The 408 upregulated S-nitrosylated peptides were subjected to consensus sequence analysis. The results showed that Lys residues in the positions from +1 to +6 and −5 to −6 were predominate around the Cys residues, which account for 298 out of 408 upregulated S-nitrosylated peptides in the *hot5-4* mutant (Fig. 2c). Interestingly, this pattern was not observed in the 379 S-nitrosylated peptides, whose abundance is relatively unaffected by the absence of *GSNOR1* (Fig. 2d). Therefore, these results suggest that Lys residues around S-nitrosylated sites may be a marker for upregulated peptides exclusively in the *hot5-4* mutant.

The results from Gene ontology (GO) analysis showed that *hot5-4* accumulated S-nitrosylated proteins are mainly involved in the oxidation-reduction processes, metabolism, transport, stress responses, development, and reproductivity (Fig. 2e). Around 82

S-nitrosylated proteins involved in stress responses, including two dehydroascorate reductases (DHAR1 and DHAR2), ascorbate peroxidase 3 (APX3), no catalyse activity1 (NCA1), nitrate reductase2 (NIA2), and pyruvate decarboxylase1 (PDC1), were upregulated in the *hot5-4* mutant. In addition, the S-nitrosylation of 21 proteins in the reproductive process involved in seed development was elevated in the *hot5-4* mutant than in the wild type. The S-nitrosylation of several proteins regulating auxin transport or signaling was also up-regulated in the *hot5-4* mutant, including ATP-binding cassette B4 (ABCB4) and ROOTS CURL IN NPA (RCN1). We noticed multiple proteins involved in plant immunity responses, including epithiospecifying senescence regulator (ESP), pectin methylesterase 12 (PME12), suppressor of SA insensitive 2 (SSI2) and calmodulin-like 24 (CML24), were also found in the list of *hot5-4* accumulated S-nitrosylated proteins. We also found that PROTOCHLOROPHYLLIDE OXIDOREDUCTASE B (PORB) was an upregulated S-nitrosylated protein in the *hot5-4* mutant determined by FAT-switch assay, which is consistent with previous nitrosoproteomics data collected by biotin-switch assay[23]. Together, the upregulated S-nitrosylated proteins involved in developmental and stress-related processes are consistent with the pleiotropic phenotypes of the *gsnor1* mutants in responses to both biotic and abiotic stresses as well as phytohormone regulation and chlorophyll metabolism[6,7,11–13,23,40,41].

We found that 12 terms of KEGG pathways were enriched in the *hot5-4* accumulated S-nitrosylated proteins (Fig. 2f and Supplementary Data 7). Interestingly, among these terms, 11 S-nitrosylated proteins are involved in protein processing in ER in Arabidopsis. With this exciting clue, we were motivated to search in literatures and we discovered that at least 34 proteins involved in ERAD and ER stress, protein glycosylation, lipid biosynthesis in ER, and ER-related transport were included in our list of S-nitrosylated proteins (Table 1). Among the 34 proteins, 17 proteins with 21 S-nitrosylation sites were overexpressed in the *hot5-4* mutant (FC > 1.2, $p < 0.05$, two-tailed unpaired $t$ test) than in the wild type. According to the results, we decided to focus on the roles of S-nitrosylation modification in regulating ER functions. which have not been reported up to the present in plants.

## S-nitrosylation of Cys337 regulates the oxidative protein-folding activity of ERO1

Previous studies have reported that ER located ENDOPLASMIC RETICULUM OXIDOREDUCTASE 1/2 (ERO1/2) are the major disulfide donors with oxidative protein-folding activity, which work with PDI and FAD to build disulfide in folding proteins[42,43]. In our proteomics data, we found that the Cys337 in ERO1 and Cys336 in ERO2 were S-nitrosylated (Figs. 3a, b), and the S-nitrosylation was upregulated in *hot5-4* than in WT (Figs. 3c, d). The S-nitrosylation of Cys346 in PDI7 is relative unchanged in *hot5-4*, when compared to that in wild type (Fig. 3c). In subsequent immunoblotting of in vitro S-nitrosylation biotin-switch assay, we confirmed that the recombinant ERO1 protein was S-nitrosylated upon incubation with GSNO. The Cys to Ser substitution at the Cys337 eliminates the S-nitrosylation of recombinant ERO1 (Fig. 3d). To detect whether S-nitrosylation at Cys337 alters the oxidase activity of ERO1, we adapted an ERO1-PDI9 oxidative protein-folding system with reduced RNase A as the substrate[44,45]. In this system, the oxidative of pre-reduced RNase A was used as the indicator of ERO1-PDI9 activity. After incubating with ERO1 proteins, RNase A reacted with 4-acetamido-4-maleimidylstilbene-2,2 disulfonic acid (AMS), which modified the free thiol groups in RNase A and led to a size increment that could be differentiated by sodium dodecyl sulfate-polyacrylamide gel electrophoresis (SDS-PAGE), since reduced RNase A with more free thiol groups runs slower after reacting with AMS than oxidized RNase A with few free thiol groups. In this experiment, the wild type and Cys337Ser mutated ERO1 proteins were pre-incubated without or with GSNO for 1 h. After removal of GSNO, the resulted ERO1 proteins then were subjected to activity assay. As the result, ERO1

pre-treated with GSNO showed an increased activity which oxidized more RNase A than untreated ERO1 (Fig. 3e, lanes 6–8 compared to 2–4, 3f and 3g). In addition, Cys337Ser mutation eliminates the effect of GSNO on ERO1 activity (Fig. 3e, lane 13–16 compared to 9–12). These results suggest that S-nitrosylation at Cys337 improves the oxidoreductase activity of ERO1.

## S-nitrosylation of Cys337 in ERO1 rearranges disulfides

It has been well documented that the oxidative activities of ERO1/ERO2 are modulated by the correct formation of disulfide bonds[45–50]. To investigate how the S-nitrosylation of Cys-337 in AtERO1 enhances its activity, we explored the disulfide bond activity in ERO1 by structure prediction and LC-MS/MS analysis. The predicted full-length structure of AtERO1 is retrieved from Alphafold Protein Structure Database (https://www.alphafold.ebi.ac.uk/) (Fig. 4a). The modeled AtERO1 forms a highly helical catalytic core and the FAD-containing reaction domain with an unstructured N-terminal region, which shows a similar topology to the structure of human Ero1α inactive form (PDB code: 3AHR)[51] (Fig. 4b). The Cys337 forms a long-range disulfide bond with Cys71, which serves as the connection between two adjacent antiparallel helixes located at opposite sides of the reaction center (Fig. 4c, upper panel). In the inactive human Ero1α, these two antiparallel helixes are connected by an anti-parallel β-hairpin, which is stabilized by two disulfide bonds Cys35-Cys48 and Cys37-Cys46 (Fig. 4c, bottom panel)[47,51]. The AtERO1 lacks a similar antiparallel β-hairpin in human ERO1α but it adapts the Cys71-Cys337 disulfide bond to stabilize the two adjacent antiparallel helixes (Fig. 4c). We then identified the disulfide bonds in recombinant AtERO1 without or with GSNO treatment by trypsin or trypsin/Glu-C digestion and subsequential LC-MS/MS analysis (Fig. 4d and Supplementary Fig. 4a–d). As the result, we identified several disulfides in the AtERO1, Cys71-Cys337, Cys222-Cys231, the inner active sites Cys375-Cys378, and Cys122/125/127-Cys151, Cys122-Cys127 in the outer active loop, which are very similar to HsEro1α (Fig. 4f, Supplementary Fig 4). Interestingly, incubation of 1 mM GSNO for 60 min strongly reduced the amount of Cys71-Cys337 disulfide bond in AtERO1 (Fig. 4e, left panel, $p = 0.004$, two-tailed unpaired $t$ test). Meanwhile, the amount of inner active site disulfide Cys375-Cys378, and Cys222-Cys231 are also reduced, while the abundance of another disulfide or non-Cys-peptide [138]KPFVPGLPSDDLK was unaffected upon GSNO treatment (Fig. 4e, Supplementary Fig. 4e). Taken the modeling structure and LC-MS/MS assay results together, the S-nitrosylation of Cys337 possibly breaks the Cys71-Cys337 disulfide bond and causes a disulfide rearrangement in AtERO1, which results in altered AtERO1 activity.

Interestingly, the sequence comparison of the ERO1 proteins from fourteen different plant species, including cotton, wheat, patens, and *Picea sitchensis* et al., revealed that Cys-71 and Cys-337 are highly conserved in diverse plant species, but are lacking in human and yeast ERO1s (Fig. 4g). Therefore, the regulating mechanism of S-nitrosylation of Cys-337 on the Cys71-Cys337 disulfide bond may be unique and highly conserved in plants.

## Altered ER functions in the *hot 5* mutants

In addition to ERO1/2, other S-nitrosylated proteins, such as FAD2, SES1, FKBP15-1/2, CRT1, CNX1, and EBS1, were also accumulated in the *gsnor1/hot5* mutant (Table 1). These proteins are known to locate in ER and have been characterized to regulate ER stress, glycosylation, or protein folding[45,52–56], strongly suggesting the critical role of protein S-nitrosylation in regulating ER functions.

By biotin-switch assay and immunoblotting, we then measured the S-nitrosylation of CTR and CNX, which have commercially available antibodies, in the wild type and the *hot5-4* mutant. The *Arabidopsis* genome encodes two CNX and three CRT homologs. The anti-CNX antibody recognizes both CNX1 and CNX2, and the anti-CRT antibody recognizes CNXs and CRTs[56]. The immunoblot results suggested that

**Table 1 | S-nitrosylated proteins involved in ER functions**

| Gene ID | Protein name | Protein function | S-nitrosylated site | Fold-change (*hot5-4*/ WS-4) | *p*-value (two-tail unpaired *t* test) |
|---|---|---|---|---|---|
| **ERAD and ER stress** | | | | | |
| AT4G24190 | HSP90.7 | An ER-resident HSP90 protein responded to tunicamycin-induced ER stress | C671 | 1.81 ± 0.39 | 0.0665 |
| AT5G61790 | CNX1 | ER chaperone protein CALNEXIN 1 | C108 | 1.5 ± 0.12 | 0.0026 |
| AT1G56340 | CRT1 | Calreticulin | C108 | 1.4 ± 0.12 | 0.0277 |
| AT2G38960 | ERO2 | Oxidoreductins | C336 | 1.4 ± 0.11 | 0.0071 |
| AT1G72280 | ERO1 | Oxidoreductins | C337 | 1.3 ± 0.03 | 0.0014 |
| AT5G48580 | FKBP15-2 | ER localized immunophilin protein which possesses PPIase activity | C45 | 1.7 ± 0.23 | 0.0077 |
| | | | C100 | 1.5 ± 0.04 | 0.0002 |
| AT3G25220 | FKBP15-1 | ER localized immunophilin | C45 | 1.1 ± 0.14 | 0.7635 |
| | | | C100 | 1.5 ± 0.04 | 0.0002 |
| AT4G29520 | SES1 | an ER localized chaperone involved in salt and heat stress response. | C128 | 0.97 ± 0.15 | 0.8078 |
| | | | C165 | 1.58 ± 0.21 | 0.0066 |
| | | | C84 | 0.21 ± 0.38 | 0.4169 |
| AT1G09310 | SVB2 | Unknown protein function in trichome formation and ER stress | C21 | 1.15 ± 0.24 | 0.4335 |
| AT4G27080 | PDI7 | protein disulfide isomerase 7 | C346 | 1.05 ± 0.13 | 0.7771 |
| AT1G17860 | KT15 | KTI family involved in negative regulation of peptidase activity | C66 | 0.54 ± 0.06 | 0.0004 |
| AT2G33120 | VAMP722 | accumulated upon ER stress | C35 | 0.84 ± 0.09 | 0.0390 |
| **Protein N-glycosylation** | | | | | |
| AT1G71220 | EBS1/UGGT | UDP-glucose:glycoprotein glucosyltransferase | C881 | 1.7 ± 0.23 | 0.0009 |
| | | | C1539 | 1.2 ± 0.04 | 0.0065 |
| AT4G21150 | HAP6/RPN2 | a subunit of OST complex involved in N-glycan biosynthesis | C76 | 1.77 ± 0.25 | 0.0084 |
| AT2G39630 | - | a dolichyl-phosphate β-glucosyltransferase in protein N-glycosylation | C253 | 1.47 ± 0.35 | 0.0968 |
| AT1G76400 | OST1b | component of the oligosaccharyl-transferase in N-glycosylation | C491 | 0.92 ± 0.16 | 0.0415 |
| AT1G20575 | DPMS1 | catalytic core of the dolichol phosphate mannase synthase | C188 | 0.91 ± 0.07 | 0.1930 |
| **Lipid biosynthesis** | | | | | |
| AT3G12120 | FAD2 | 18:2 fatty acid synthesis in ER | C26 | 3.0 ± 1.61 | 0.0244 |
| AT1G27950 | LTPG1 | a lipid transfer protein located in ER | C45 | 1.31 ± 0.02 | 0.0000 |
| AT5G13640 | PDAT1 | Hospholipid:Diacylglycerol Acyltransferase | C129 | 1.10 ± 0.14 | 0.3721 |
| AT4G24510 | CER2/VC2 | C28 to C30 fatty acid elongation | C287 | 1.02 ± 0.07 | 0.7098 |
| AT4G36480 | LCB1 | subunit of serine palmitoyltransferase | C132 | 0.99 ± 0.10 | 0.8358 |
| AT3G57650 | LPAT2 | an ER localized protein involved in lipid biosynthesis | C89 | 0.92 ± 0.07 | 0.4781 |
| AT1G67730 | KCR1/YBR159 | BETA-KETOACYL REDUCTASE 1 in very long chain fatty acid elongation | C233 | 0.80 ± 0.11 | 0.0538 |
| **ER-related transport** | | | | | |
| AT2G01470 | SEC12 | involved in ER and Golgi transport | C270 | 1.25 ± 0.16 | 0.0407 |
| AT2G20990 | SYT1 | Synaptotagmin in ER-PM boundary | C145 | 0.91 ± 0.03 | 0.0968 |
| AT3G07680 | P24β2 | p24 protein in ER and Golgi transport | C31 | 1.41 ± 0.20 | 0.0023 |
| AT3G09740 | SYP71 | ER located Qc-SNARE protein | C14 | 1.15 ± 0.31 | 0.6338 |
| AT3G52850 | VSR1/BP80 | vacuolar sorting receptor | C202 | 1.20 ± 0.11 | 0.0979 |
| | | | C257 | 1.79 ± 0.10 | 0.0035 |
| | | | C319 | 1.46 ± 0.08 | 0.0001 |
| AT4G34450 | γ2-cop-3 | a component of Coat Protein I complex (COP1) in Golgi-ER transport | C46 | 0.69 ± 0.13 | 0.1233 |
| AT3G63460 | SEC31B | a component of the COPII | C339 | 1.50 ± 0.08 | 0.0043 |
| | | | C584 | 1.94 ± 0.29 | 0.0104 |
| | | | C738 | 0.96 ± 0.14 | 0.5550 |
| AT4G14160 | SEC23F | a component of the COPII | C432 | 1.41 ± 0.05 | 0.0005 |
| **Other ER-related processes** | | | | | |
| AT3G09260 | LEB | ER body β-glucosidase | C222 | 0.42 ± 0.05 | 0.0069 |
| AT4G37640 | ACA2 | ER-located calmodulin-regulated $Ca^{2+}$-pump | C28 | 1.11 ± 0.06 | 0.0597 |
| | | | C669 | 1.02 ± 0.10 | 0.8237 |

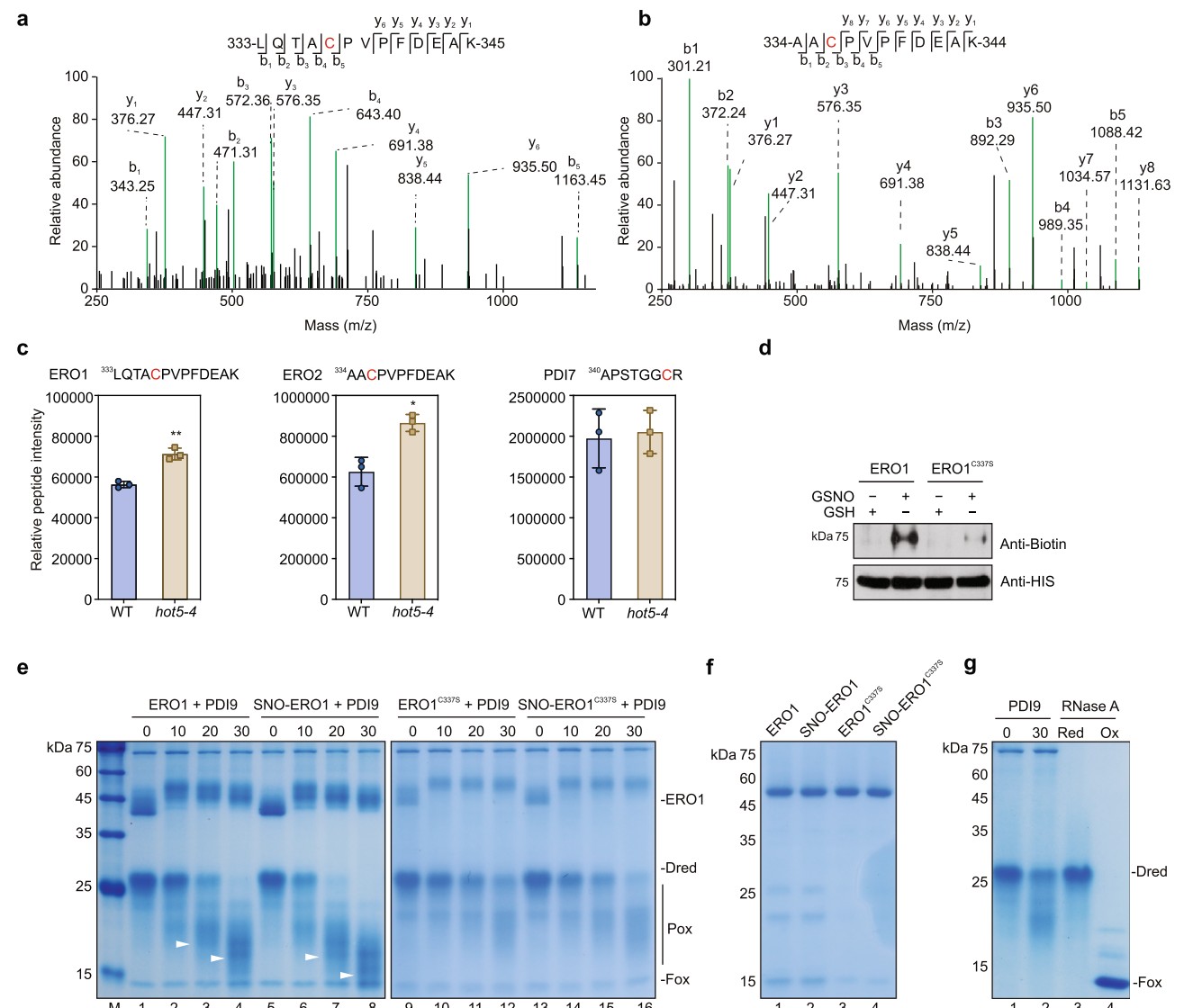

**Fig. 3 | S-nitrosylation of Cys337 in ERO1 enhance ERO1 activity. a** The MS/MS spectrum of S-nitrosylated peptide containing Cys337 in ERO1. **b** The MS/MS spectrum of S-nitrosylated peptide containing Cys336 in ERO2. **c** The relative intensity of S-nitrosylated peptides containing Cys337 in ERO1, Cys336 in ERO2, and Cys346 in PDI7, in wild type and *hot5-4* mutant. The S-nitrosylated Cys residues are highlighted in red. Error bars, SD (*n* = 3 biological replicates). * *p* < 0.05, ** *p* < 0.01, two-tailed unpaired *t* test. **d** Analysis of S-nitrosylation of recombinant wild-type and Cys337Ser mutated ERO1 without or with GSNO treatment. The S-nitrosylated recombinant HIS-ERO1 was detected by anti-biotin immunoblot and the loading of HIS-ERO1 and HIS-ERO1^C337S was measured by anti-HIS immunoblot, respectively. Images shown are representative of at least two independent experiments.

**e** Analysis of ERO1 and ERO1^C337S activities without or with GSNO treatment by gel-based RNase A refolding assay. Images shown are representative of at least three independent experiments. **f** The reduced SDS-PAGE result show the amount of wild-type and Cys337Ser mutated recombinant ERO1 proteins used in (**e**). Images shown are representative of at least three independent experiments. **g** The control reactions without AtERO1. RNase A was separated by 15% nonreducing sodium dodecyl sulfate-polyacrylamide gel electrophoresis (SDS-PAGE) followed by Coomassie Blue staining. Dred: denatured and reduced RNase A; Pox: partially oxidized RNase A; Fox: fully oxidation RNase A. Images shown are representative of at least two independent experiments.

the S-nitrosylation in CRT and CNX proteins was upregulated in the *hot5-4* mutant than in the wild type (Fig. 5a).

Encouraged by the results that multiple key enzymes in ER functions undergo S-nitrosylation, we then measured the responses of the wild type and the *hot5* mutants to tunicamycin (Tm), an ER stress inducer. The addition of 0.05 μg/mL Tm strongly inhibited the growth and seedling establishment in the Col and Ws wild type seedlings, in the context of fresh weight (75.4% reduction, *p* < 0.05, two-tailed unpaired *t* tests) and chlorophyll content (37.9% reduction, two-tailed unpaired *t* tests, *p* < 0.05) (Fig. 5b–d and Supplementary Fig. 5). Interestingly, both the *hot5-2* (Col) and *hot5-4* (Ws) alleles showed strong resistance to Tm, and the growth and chlorophyll content of the *hot5* mutants under Tm conditions was comparable to

the content under the control conditions (two-tailed unpaired *t* tests, *p* < 0.05). We also measured the expression of unfolded protein response (UPR) marker genes in the wide type and the *hot5-4* mutant with or without Tm treatment. Consistent with previous studies, all tested UPR marker genes, *BiP3*, *BiP1,2*, *ERDJ3A*, and *SHD*, showed a Tm-induced expression in both the wild type and the *hot5-4* mutant seedlings (Fig. 5e–h). However, these genes, besides *BiP3*, showed an enhanced expression in the *hot5-4* mutant, even without Tm treatment (two-tailed unpaired *t* tests, *p* < 0.05). Consistent with the Tm-insensitive phenotype, the Tm-induced expression of these genes (fold-change) was impaired by the deficiency of *GSNOR1* (wild type vs. *hot5-4*). These results suggested that ER function is altered in the *hot5-4* mutant, resulting in resistance to Tm treatment.

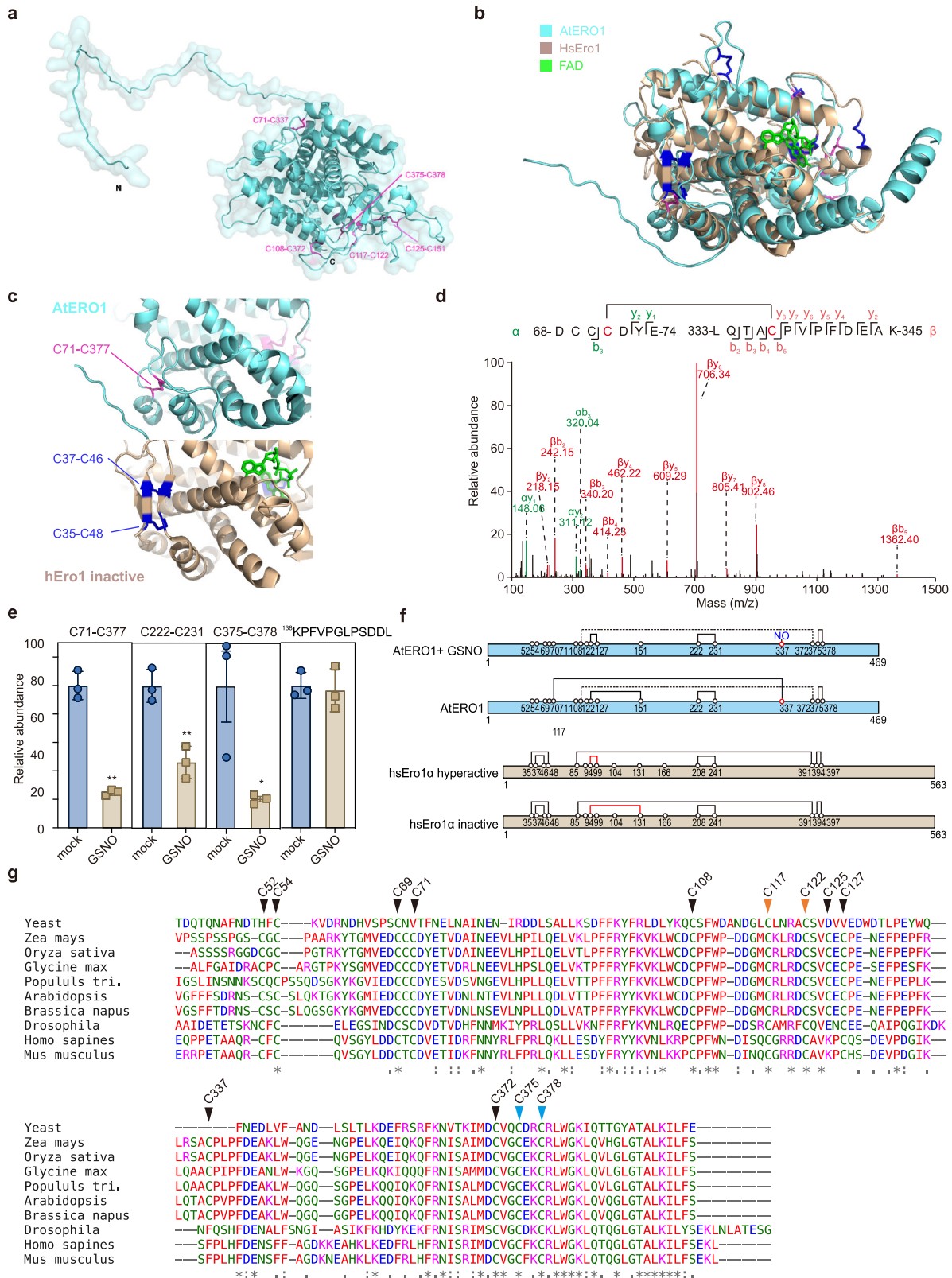

## Discussion

S-nitrosylation is a reversible redox-based modification that controls various developmental processes and environmental responses in bacteria, plants, and animals. Due to the relatively low abundance and high dynamics of S-nitrosylated cysteine, it is challenging to quantitively profile the global S-nitrosylation in vivo with high sensitivity and efficiency. In this study, we used a FAT tag, instead of the traditional biotin-switch method, to label the S-nitrosylated cysteine residue, which allowed for an in-depth comprehensive analysis of the S-nitrosylation at the proteome level in the model plant Arabidopsis. Compared to the classic biotin-switch method, the FAT-switch method has several advantages, including improved sensitivity and efficiency and comparable specificity, partly due to that the FAT-switch method reduces the complexity of fragmentation and increases the intensity of S-nitrosylated peptides

**Fig. 4 | Application of GSNO affect the disulfide bonds distribution in AtERO1.**
**a** Structural modeling of AtERO1, with disulfide bonds shown by purple sticks.
**b** Superimposed structure of modeled AtERO1 (cyan) with the inactive form of
hsEro1α (brown, PDB code: 3AHR). Inset highlights the close-up view of Cys71-
Cys337 disulfide bond in AtERO1 (purple sticks), and the disulfide bridges formed in
the same region of hsEro1α (blue sticks). Green sticks represent the FAD molecule.
**c** Comparison of the disulfide bonds in N-terminal of AtERO1 and hsEro1α. The
disulfides Cys35-Cys48 and C37-Cys46 in hsEro1α, and Cys71-Cys337 in AtERO1 are
highlighted. **d** The MS/MS spectrum of Cys71-Cys337 disulfide bond identified in
AtERO1. The α and β indicates two peptides contain Cys71 and Cys337, respectively.
**e** The relative intensity of Cys71-Cys337, Cys222-Cys231, Cys375-Cys378 disulfide in
ERO1 protein without or with incubation of GSNO. The trypsin/Glu-C digested

AtERO1 peptides were subjected into LC-MS/MS analysis, the relative intensity of
disulfide peptides and [138]KPFVPGLPSDDL were obtained from the result of pLink
and Proteome Discovery analyses, respectively. Error bars, SD ($n = 3$ biological
replicates). *$p < 0.05$, **$p < 0.01$, two-tail unpaired $t$ test. **f** Schematic representation
of cysteine positions in AtERO1 and hsEro1α. Lines indicate disulfide bonds. The
Cys108-Cys372 disulfide that not detected by LC-MS/MS is indicated by dot line.
**g** Alignment of ERO1 protein sequences in different model species. The alignment
was performed with MUSCLE (https://www.ebi.ac.uk/Tools/msa/muscle). The
cysteine positions in AtERO1 are indicated by black arrows, and the residues cor-
responding to outer and inner active sites are highlighted by red and blue arrows,
respectively.

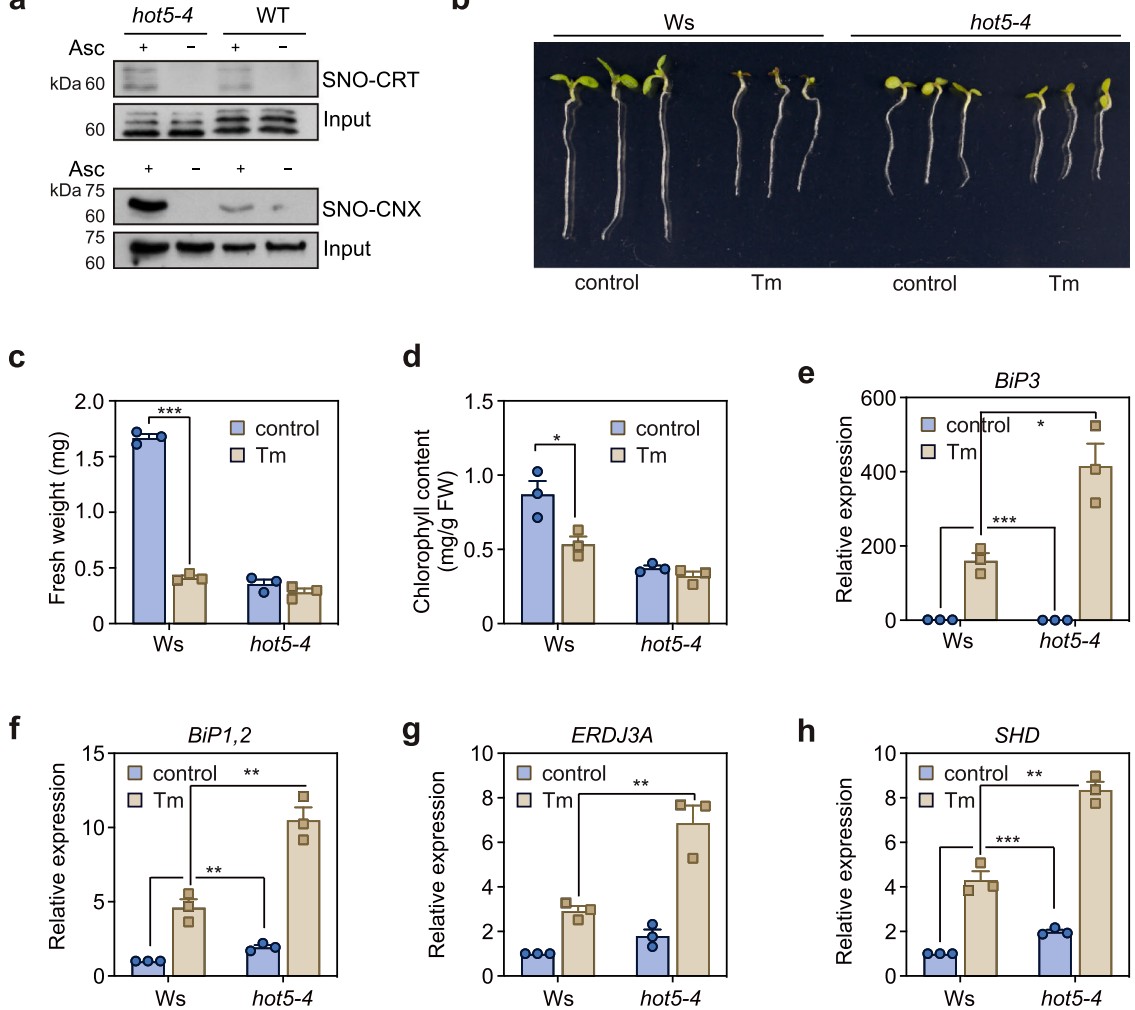

**Fig. 5 | The *hot5-4* mutant shows the tolerance phenotype of ER stress under Tm
treatment. a** Analysis of S-nitrosylation of CNX and CRT proteins in wild type and
*hot5-4* seedlings by in vivo biotin-switch method. The blots were detected by the
anti-CRT and anti-CNX antibody, respectively. Images shown are representative of
at least two independent experiments. **b** Phenotype of wild type (WT) and *hot5-4*
with or without Tm treatment. Photograph were taken after growing at 22 °C for
9 days. **c** Fresh weight of WT and *hot5-4* plants with or without Tm treatment. Error

bars, SD ($n = 3$ biological represents and each replicate consisting of 10 seedlings).
***$p < 0.001$, two-tail unpaired $t$ test. **d** Chlorophyll content of WT and *hot5-4* plants
with or without Tm treatment. Error bars, SD (($n = 3$ biological represents and each
replicate consisting of 10 seedlings). *$p < 0.05$. two-tailed unpaired $t$ test.
**e**–**h** Expression of *BiP3*, *BiP1,2*, *ERDJ3A* and *SHD* in wild type (WT) and *hot5-4* with or
without Tm treatment. Error bars, SD ($n = 3$ biological replicates). *$p < 0.05$,
**$p < 0.01$, ***$p < 0.001$, two-tailed unpaired $t$ test.

in MS/MS. As far as we know, the sensitivity and efficiency of the FAF-
switch method are better than other reported alternative approaches
used in plants. Compared to dozens to hundreds S-nitrosylated pro-
teins/peptides identified by biotin-switch or iodoTMT methods in
plants (Supplementary Table 1), the FAT-switch method identified up to
2,500 S-nitrosylation peptides in a single run of LC-MS/MS assay,
making it a very powerful tool to measure the dynamics of

S-nitrosylation in response to stimuli. The efficiency and sensitivity of
our FAT-switch method are even comparable to the Cys-BOOST
method[29], the most efficient S-nitrosoproteomics method reported in
animal cells by far, in the context of identified S-nitrosylation sites
(2,086 vs. 2,151 sites). Notably, we identified unique S-nitrosylated
peptides by the FAT-switch method compared to the biotin-switch
method, likely due to the FAT-switch method prefers to enrich the

hydrophobic peptides. In conclusion, combining these two methods might significantly enhance the coverage of S-nitrosoproteome and will allow for the acquisition of deeper global S-nitrosylation profiles. However, one limitation is that only 25.6% to 30.6% of the enriched peptides are S-nitrosylated peptides, which leaves room for further improvement. In addition, as lacking commercial antibody recognizing F-tag, the FAT-switch enriched proteins couldn't not be subjected into immunoblot, limits its application beyond proteomics study.

More importantly, the FAT-switch method allowed us to perform the in-depth quantitative analysis of S-nitrosoproteome in plants. To our knowledge, this is the first large-scale quantitative S-nitrosoproteome study with such an extensive coverage in plants. The increment of global S-nitrosylation in the *hot5-4* mutant is consistent with the high concentration of endogenous GSNO and the increased overall protein S-nitrosylation observed by immunoblot in the *hot5-4* mutant. Identification of enriched S-nitrosylated proteins in the *hot5-4* mutant will provide important clues on the mechanisms by which elevated endogenous GSNO concentration regulates plant growth, development, reproduction, and response to abiotic stress and pathogenies. The quantitative S-nitrosoproteomics revealed that an auxin transport, ATP-BINDING CASSETTE B4 (ABCB4), and three other proteins involved in auxin signaling showed an upregulated S-nitrosylation in the *gsnor1* mutant. The changes of these direct S-nitrosylation targets might lead to the auxin-related morphological phenotypes in the *gsnor1/hot5* mutant. In addition to auxin signaling, several proteins involved in the developmental and morphological processes also showed increased S-nitrosylation in the *gsnor1/hot5* mutant than in the wild type. AtPER1, a 1-CYSTEINE PEROXIREDOXIN, known to enhance seed dormancy and reduce seed germination, is a potential S-nitrosylation target that regulates germination along with ABI5 and SnRK2s. Several upregulated S-nitrosylated proteins in the *gsnor1/hot5* mutant, like HSP90.2, ESP, and HEN2, are known to function in immunity and heat stress, consistent with the heat stress sensitivity and the impaired immune response observed in the *gsnor1/hot5/par2* mutants[4,5,57]. Additionally, identification of hundreds of additional S-nitrosylation targets provides valuable resources for studying the biological function of S-nitrosylation in plants.

Based on the proteomics results, we further revealed an unreported function of protein S-nitrosylation in regulating ER functions in plants. The ER is a major protein folding compartment for secreted, plasma membrane and organelle proteins. Roughly one-third of all eukaryotic proteins are folded and maturated in ER. In general, newly synthesized proteins translocate from the cytosol into ER in an unfolded state, where ER-resident molecular chaperones and folding catalysts promote protein folding to acquire their appropriate conformations. Because protein folding is highly prone to errors, all eukaryotic cells have evolved specific protein quality-control processes that include the promotion of protein folding and recognition of misfolded proteins for degradation[58,59]. Physiological demands and environmental stresses increase the protein processing loading and impact the ER homeostasis in plants, resulting in the accumulation of unfolded or misfolded proteins in the ER lumen, leading to ER stress. Under normal conditions, ER chaperone BiPs bind to the ER luminal domain of three stress sensors on ER membrane, Ire1, ATF6, and PERK. While upon ER stress, BiPs dissociate from ER stress sensors and bind to the abnormal proteins. Dissociation of BiPs also activates ER stress sensors and triggers ER-stress responses, including translational attenuation, induction of ER chaperones such as BiPs to enhance protein folding, degradation of misfolded proteins through ubiquitin proteasome system in ER, and activation of apoptosis to remove damaged cells. In this study, we identified several ER chaperones and key enzymes in protein glycosylation, folding, transport, and lipid biosynthesis which undergo S-nitrosylation (Table 1). Consistently, GSNOR1 is known to locate in ER[60], and dysfunction of GSNOR1 in the *gsnor1/hot5* mutants leads to the increased expression of ER-stress

responsive genes, like *BiPs*, *ERDJ3A*, and *SHD*, resulting in an insensitive response to Tm-induced ER stress. The mutants that lack S-nitrosylation target proteins, such as *crt1a* or *ebs1*, also showed altered sensitivity to Tm[61,62]. These results suggested an unreported, crucial function of NO/S-nitrosylation in regulating ER function in plants, and, more importantly, revealed putative functional S-nitrosylation targets in ER. In mammals, a few studies showed that NO-dependent protein modification is involved in ER functions. NO accumulation disturbs the $Ca^{2+}$ homeostasis and induces ER stress responses[63,64], likely through tyrosine nitration of SERCA, and poly S-nitrosylation of cardiac calcium release channels[65,66]. In brains manifesting sporadic Parkinson's or Alzheimer's disease, NO induces S-nitrosylation of PDI, which inhibits its enzymatic activity, leading to the accumulation of polyubiquitinated proteins and activation of the unfolded protein response[67]. Recent studies showed that S-nitrosylation of a key UPR regulator, IRE1α impairs ER function and prolongs ER stress in obesity[68,69]. The discovery of novel S-nitrosylation targets in ER will reveal the molecular mechanism underlying ER stress responses and provide novel candidates to improve ER function and homeostasis upon stress or disease conditions in the future.

ERO-PDI are major molecular chaperones and folding catalysts in ER, and over 30% of proteins require the chaperone PDI to promote disulfide bond formation. The ERO1 catalyzes the disulfide bond formation in nascent polypeptide substrates via electron transfer through PDI with oxygen as the final electron acceptor[70]. Dysfunction of ERO1/ERO2 or multiple PDIs in Arabidopsis resulted in hypersensitivity to DTT induced ER stress[45,71]. The double mutant *ero1ero2* is extremely sensitive to reductive stress and hypoxia[72]. In this study, we discovered that the S-nitrosylation of ERO1/ERO2 was overexpressed in the *gsnor1/hot5* mutant. Interestingly, the S-nitrosylation at Cys337 in ERO1 positively regulates its activity, suggesting a crucial role of NO/nitrosylation in regulating of ER redox state. It has been documented that the *gsnor1/hot5* mutants were tolerant to oxidative stress induced by paraquat, a nonselective herbicide[7,73]. ER stress causes oxidative damage through increased $H_2O_2$ accumulation which triggers ROS signaling and changes the redox state[74]. Upon stress condition, the endogenous NO elevates ERO1 activity by S-nitrosylation, and therefore promotes disulfide bond formation and protein folding and reduces the oxidative damage which potentially contributes to the oxidative tolerance of the *gsnor1/hot5* mutant. Besides ERO1, the S-nitrosylation of other redox enzymes, such as GSH1, PDI7, PER1, and APX3, are also overexpressed in the *gsnor1/hot5* mutant than in the wild type. The specific crosstalk between ROS- and NO-signaling to damping oxidative damage in ER is another possible explanation of the resistance to ER and oxidative stresses in the *gsnor1/hot5* mutant.

It is well-documented that the ERO1/2 oxidoreductase activity is regulated by different disulfide forming. Human Ero1α is regulated via the rearrangement of the disulfide bond between shuttle Cys-94 and shuttle Cys-99 with noncatalytic Cys-131 and noncatalytic Cys-104 in the regulatory loop[46,75,76]. The Cys94-Cys131 disulfide blocks the shuttle Cys94, results in inactivation of HsEro1α[51,77]. Cys104Ala and Cys131Ala mutant HsEro1α that homogeneously forms the Cys94-Cys99 disulfide is hyperactive[78]. Beside the outer activation site, mutating the inner active site Cys394 or Cys397 also impaired HsEro1α activity[76]. The modeling AtERO1 structure and LC-MS/MS results suggested the similar disulfide distribution in AtERO1 and HsEro1α in (Fig. 4). Pervious study showed that mutation Cys177/Cys122 (corresponding to Cys108 and Cys94 in HsEro1α) or Cys375/Cys378 (corresponding to Cys394 and Cys397 in HsEro1α) abolished AtERO1 activity[45]. In this study, we showed that GSNO treatment, which breaks the long-range disulfide bond Cys71-Cys337 in AtERO1, resulted in the alteration of disulfide distribution and oxidoreductase activity (Figs. 3d and 4). The S-nitrosylation of Cys337 may serve as a critical regulation mechanism for ERO1 activity, which could coordinate the endogenous SNO status

with ERO1 activity. Interestingly, the S-nitrosylation dependent ERO activity regulation may be only conserved in plants. The animal ERO1s do not contain this cysteine residue corresponding to Cys337 in AtERO1 (Fig. 4g), and removal of the N-terminal region containing conserved cysteine residue corresponding to AtERO1 Cys71 likely did not inhibit HsEro1α oxidative activity[47,51]. Thus, the plant's ERO proteins may have unique structure and regulation mechanism, which differ from their mammalian/yeast homologies. However, as high enrichment of cysteine and negative-charged residues, we cannot clearly detect the disulfide bonds between the shuttle Cys residues and nearby noncatalytic Cys residues in the flexible regulatory loop. In the future, it would be of great interest to explore how S-nitrosylation of Cys337 affects the arrangement of intramolecular disulfide and underlies the molecular mechanism of ERO1 activity regulation. Additionally, we would like to note that while we used ERO1/2 as a case study to validate our FAT-switch result, S-nitrosylation is known to target multiple proteins related to ER function in plants (Table 1). Therefore, it would be interesting to further investigate the role of other S-nitrosylation targets in the ER and explore how S-nitrosylation affects their function in ER processes. Such investigation would provide a more complete understanding of how S-nitrosylation regulates ER function in plants.

## Methods

### Plant materials

The *Arabidopsis* wild type Ws/Col-0 and mutant *hot5-4/hot5-2* seeds were sterilized with 70% ethanol solution, and then seeds were germinated in 40 ml 1/2 MS liquid medium containing 0.5% (w/v) sucrose after 2-day 4 °C treatment. Seedlings were grown at 22 °C with 16 h light for 10 d before being harvested for S-nitrosoproteomics assay. For the phenotypic assay, wild type and *hot5* mutant seeds were germinated in 0.5 ml 1/2 MS liquid medium containing 0.5% sucrose with or without 0.05 µg/ml Tm. The fresh weight and chlorophyll content were measured 9 d after germination. Total chlorophyll was extracted from seedlings and analyzed as described[79].

### In vivo S-nitrosylated protein enrichment

0.5 g seedling samples were homogenized in 1.5 ml of ice-cold HEN buffer (250 mM HEPES, pH 7.7, 1 mM EDTA, 0.1 mM neocuproine) containing 0.5% Triton X-100, 0.1 mM PMSF and protease inhibitor cocktail for 30 min with shake. The protein extracts were centrifugated at 20,000 × g for 15 min at 4 °C, and the supernatant was collected. The proteins were diluted to 1 mg/ml by HEN buffer and were incubated in 3 volumes of blocking buffer (250 mM HEPES, pH 7.7, 1 mM EDTA, 0.1 mM neocuproine, 5% SDS, 270 mM IAM) at 37 °C for 120 min. The samples were precipitated by incubating with 2 volumes of acetone at −20 °C for 30 min. The pellet was washed three times with 70% acetone and dissolved in 0.5 ml of HEN buffer with 1% SDS. The concentration of proteins was detected by BCA assay, and 12.5 µl of 200 mM sodium ascorbate and 200 µg of TFIA were added to samples which contained 3 mg proteins in 0.5 ml of HENS buffer. After labeling for 2 h at room temperature with shaking and centrifuging at 200 × g, the supernatant was collected; extra TFIA was removed by 10 kDa filter. Proteins were digested by the FASP method (10 mM DTT treatment for 90 min and 55 mM IAM treatment for 25 min, trypsin 1:50, digestion overnight). The samples were dried in a centrifugal vacuum concentrator (Labconco, USA).

For enrichment by the C18 column, dried peptides were re-dissolved in 800 µl 20% ACN with 0.1%TFA. Sep-Pak 100 mg C18 columns (Waters, USA) were washed sequentially by 1 ml methanol, 1 ml 80% ACN (containing 0.1% TFA) and 1 ml 0.1% TFA. After samples were loaded to C18 columns, 500 µl 20% ACN (containing 0.1% TFA) was added to remove hydrophilic and none TFIA-tag peptides. 1 ml methanol was added to elute the peptides. The samples were dried in a

centrifugal vacuum concentrator (Labconco, USA) and re-dissolved in 200 µl 50% methanol/50% 25 mM ammonium bicarbonate (ABC).

A home-made nanographite column was used to enrich TFIA-tag peptides. Firstly, 5 mg nanographite Fluoride particles (XFNANO, China) were packed into 200 µl tip. 200 µl methanol and 50% methanol/50% 25 mM ABC were used to wash the column by centrifuging at 200 × g. The column was loaded with 200 µl sample (50% methanol/50% 25 mM ABC) and was washed sequentially by 200 µl 25 mM ABC, 200 µl 20% methanol/80% 25 mM ABC, 200 µl 40% methanol/60% 25 mM ABC and 200 µl 60% methanol/40% 25 mM ABC. Finally, 100 µl methanol was added to elute the TFIA-tag peptides. The eluted samples were dried in a centrifugal vacuum concentrator (Labconco, USA) and redissolved in 6 µl 0.2% FA (containing 5% ACN) solution for LC-MS/MS analysis.

The biotin switch assay was performed as described previously[22,23].1 g seedling samples were homogenized in 3 ml of ice-cold HEN buffer (250 mM HEPES, pH 7.7, 1 mM EDTA, 0.1 mM neocuproine) containing 0.5% Triton X-100, 0.1 mM PMSF and protease inhibitor cocktail for 30 min with shake. The protein extracts were centrifugated at 20,000 × g for 15 min at 4 °C, and the supernatant was collected. The proteins were diluted to 1 mg/ml by HEN buffer and were incubated in 3 volumes of blocking buffer (250 mM HEPES, pH 7.7, 1 mM EDTA, 0.1 mM neocuproine, 5% SDS, 270 mM IAM) at 37 °C for 120 min. The samples were precipitated by incubating with 2 volumes of acetone at −20 °C for 30 min. The pellet was washed three times with 70% acetone and dissolved in 0.5 ml of HEN buffer with 1% SDS. The concentration of proteins was detected by BCA assay, and 12.5 µl of 200 mM sodium ascorbate and 1 mg of Biotin-HPDP (solubility in DMSO) were added to samples. After labeling for 2 h at room temperature with shaking and centrifuging at 200 × g, the supernatant was collected; extra Biotin-HPDP was removed by 10 kDa filter. Proteins were digested by the FASP method (10 mM DTT treatment for 90 min and 55 mM IAM treatment for 25 min, trypsin 1:50, digestion overnight). The samples were dried in a centrifugal vacuum concentrator (Labconco, USA). After tryptic digestion, NeutrAvidin agarose resins (Thermo Scientific) were used for purification of the S-nitrosylated peptides by affinity chromatography. The purified peptides were desalted and then subjected to LC-MS/MS analysis.

### TMT labeling

A TMT sixplex was used to label peptides for quantitative analysis. Digested peptides (200 µg) were resuspended in 100 µl of 200 mM HEPES containing 20% ACN, pH 8.5. TMT-126, 127, 128, 129, 130, 131 reagents were dissolved in 40 µl of anhydrous ACN and vortexed for 5 min. 40 µl of TMT reagents were added into six samples and incubated for 1 h. The reaction was quenched by adding 8 µl of 5% hydroxylamine and incubating for 15 min. All six labeled peptide samples (126-131) were mixed into a pool in a 1.5 ml Eppendorf tube. Labeled peptides were acidified by adding 20 µl of FA to pH=3.0. The samples were dried in a centrifugal vacuum concentrator (Labconco, USA).

### LC-MS/MS assay

Peptide samples were dissolved in 6 µl 0.1% formic acid and 4 µl was injected into an EASY-nLC 1200 HPLC system (Thermo Fisher Scientific). Eluent was introduced into the mass spectrometer, using a 15 cm in-house C18 capillary column (Polymicro; 75 µm inner diameter) packed with 2.2 µm C18 resin (Bischoff Chromatography). The mobile phase buffer consists of 0.1% formic acid in water (buffer A) with an eluting buffer of 0.1% formic acid in 80% (v/v) acetonitrile (buffer B). The gradient was set as 10-70% buffer B for 100 mins and 70-90% for 10 mins at 300 nl/min. This LC system was online with an Orbitrap Fusion mass spectrometry (Thermo Fisher Scientific) using high energy collision-induced dissociation (HCD) mode. MS survey scan was performed at a resolution of 120,000 over the *m/z* range of 300−1500, and MS/MS was selected by data-dependent scanning on

Top-Speed mode at 15,000 resolution (automatic gain control (AGC) 1E5, maximum injection time 120 ms). Dynamic exclusion was set to a period of 60 s. Data acquisition was controlled by Xcalibur (version 3.0, Thermo Fisher Scientific).

## Mass spectrometry data analysis

MS/MS data were searched against the Arabidopsis Information Resource (TAIR 10) protein sequences database using Proteome Discoverer 2.2 (Thermo Fisher Scientific). The search parameters were restricted to tryptic peptides at a maximum of two missed cleavages. Cysteine carbamidomethylation, oxidation of methionine and TFIA (C11F13H9NO, 418.047656 Da) of cysteine were set as variable modifications. The minimum peptide length was set at six amino acids. Mass tolerances were set up to 10 ppm for MS ions and 0.1 Da for MS/MS fragment ions. False discovery rate (FDR) of peptides was set up to less than 1%. For biotin switch assay, cysteine biotinylation was added as one of variable modifications. For TMT quantitative analysis, TMT6plex of lysine or N-terminal of peptides was added as one of variable modifications.

## Label-free quantification for total proteins

Three biological replicates of wild-type and mutant seedlings were prepared. The protein extraction and digestion were performed as described previously[80]. The samples were grounded into powder with liquid nitrogen in mortar and lysed in GdmCl lysis buffer (6 M guanidine hydrochloride, 100 mM Tris-Cl, pH 8.5, 10 mM TECP, 40 mM 2-chloroacetamide). Lysates were boiled at 95 °C for 5 min and sonicated three times, 10 sec each time. After centrifugation, proteins in the supernatants were precipitated by methanol-chloroform. The precipitated protein pellets were air dired and resuspend in 100 µL of digestion buffer (12 mM sodium deoxycholate(SDC), 12 mM sodium lauroyl sarcosinate(SLS), 100 mM Tris-Cl, pH 8.5). Protein amount was quantified by BCA kit.

Protein extracts were 10-fold diluted by TEAB (triethylammonium bicarbonate) to dilute the digestion buffer. Proteins were then digested by Lys-C (FUJIFILM Wako Chemicals) at 37 °C for 3 h,and by trypsin (Promega) at a final 1:100 (w/w) enzyme to sample ratio at 37 °C overnight. The SDC and SLS were separated from digested peptides by acidifying the solution using 10% TFA. The digests were then desalted using a 100 mg Sep-Pak C18 column (Waters).

After quantified by BCA kit, the total peptides were resuspended in 0.1% (v/v) formic acid solution, and then analyzed by an Orbitrap Fusion mass spectrometry (Thermo Fisher Scientific). The peptides were separated by Easy-nLC 1200 (Thermo Fisher Scientific), equipped with 150 mm in-house C18 analytical column. The peptide mixture was separated on the analytical column with the following conditions at 300 nl/min, 210 min run with a liquid gradient from 5–40% of solvent B (80% acetonitrile/0.1% FA). The parameters for a full MS survey scan were set as a resolution of 60,000 at 400 $m/z$ over the $m/z$ range of 300–1500, Automatic gain controls (AGC) target of 2E5, maximum ion injection time (IT) of 60 ms. The top 20 multiply-charged parent ions were selected by data dependent MS/MS mode, fragmented by the high energy collision-induced dissociation (HCD). For the MS/MS detection, the resolution was set at 15,000, AGC target value was 1E5, and the maximum IT was 120 ms. Dynamic exclusion was enabled for 60 s. Data acquisition was controlled by Xcalibur (version 3.0, Thermo Fisher Scientific).

The raw files were submitted to Proteome Discoverer software (Thermo Fisher Scientific, version 2.2) for peptide identification and label-free quantitation and searched against the Arabidopsis Information Resource (TAIR 10). The following parameters were used: carbamidomethylation of Cys was set as a fixed modification, oxidation of Met was set as variable modifications, and a maximum of two missed cleavage was allowed. The minimum peptide length was set at six amino acids. Mass tolerances were set up to 10 ppm for MS ions and 0.1 Da for MS/MS fragment ions. The false discovery rates of peptides were set at 1% FDR.

## Gene ontology analysis

Consensus sequences of S-nitrosylated peptides were predicted using the pLogo method[39]. GO analysis was performed using DAVID Bioinformatics Resources[81]. Principal component analysis (PCA) of all samples was performed by ClustVis[82]. Protein Properties Analyses Software (ProPAS) were used to calculate the hydrophobicity values for SNO peptides[83].

## Immunoblot assay

For immunoblotting, 100 µl (1 µg/µl) IAM blocked proteins, 1 µl of 1 M sodium ascorbate and 5 mM of Biotin-maleimide (100 mM, solubility in DMSO) were mixed in a 1.5 ml tube. After labeling for 90 min at room temperature with shaking, the proteins were washed two times with neutralization buffer (25 mM Hepes, pH 7.7, 100 mM NaCl, 1 mM EDTA, 0.5% Triton X-100) in 10 kDa filter to remove Biotin-maleimide. The proteins were resolubilized in 1 ml neutralization buffer, and incubated with 30 µl of streptavidin beads at 4 °C overnight. After the incubation, the beads were washed five times with wash buffer (25 mM HEPES, pH 7.7, 600 mM NaCl, 1 mM EDTA, 0.5% Triton X-100) and finally washed with 1 ml PBS. The proteins were then eluted by boiling in 30 µl of 2 × SDS sample buffer. All samples were analyzed in 12% SDS-PAGE and immunoblotting by indicated antibodies. Primary antibodies used in this study are as follows: anti-RPN12a (Agrisera, AS194268), anti-SDIRIP1 (Agrisera, AS132729), anti-PSB33 (Agrisera, AS121852), anti-CNX (Agrisera, AS122365), and anti-CRT from Jianming Li's Lab.

For in vitro S-nitrosylation assay by immunoblotting, approximately 100 µg recombinant protein was incubated with 1 mM GSNO or GSH at room temperature in the dark for 60 min, and excess GSNO or GSH was removed using 10 kDa filter. The sample was blocked as described above and dissolved in 100 µl of HENS buffer, followed by adding 5 µl of 200 mM sodium ascorbate and 25 µl of 4 mM biotin-HPDP. After labeling for 1 h at room temperature, the sample was directly separated by SDS-PAGE without boiling and analyzed by immunoblotting using anti-biotin antibody.

## Measurement of gene expression

Total RNA was extracted from *Ws* and *hot5-4* seedlings. Total RNA was isolated using the RNeasy Plant Mini Kit (QIAGEN) according to the manufacturer's instructions. For real-time PCR assays, reactions were set up with iQ SYBR Green Supermix (Bio-Rad). A CFX96 Touch Real-Time PCR Detection System (Bio-Rad) was used to detect amplification levels. Quantification was performed using three independent biological replicates. The primers used in real-time PCR were listed in Supplementary Table 2.

## Protein expression and purification

For recombinant protein expression, the coding sequence of PYL4, PDI9, and ERO1 were subcloned into pRSF-Duet-1 vector to fuse a MSB tag to the N-terminus of the target protein. The plasmids were transformed into *Escherichia coli* strain BL21 (DE3) RIL (Stratagene). Expression was induced at 16 °C overnight with 0.2 mM of IPTG. The recombinant proteins were purified with a prepackaged His-Trap FF column (GE Healthcare) and further purified using a Q FF column (GE Healthcare). The MSB tag was removed by TEV protease (Yeasen) before used in further activity assay. The primers used in vector construction were listed in Supplementary Table 2.

## RNase A reoxidation analysis

To denature RNase A, 2 mg RNase A was added into 1 ml 8 M urea containing 150 mm DTT. After 2 h shaking, urea and DTT were replaced by 100 Tris-HAc, pH 8.0, 50 mM NaCl, and 1 mM EDTA. For GSNO incubation, 100 µg recombinant wild type or Cys337 mutated ERO1

proteins were incubated with 2.5 mM GSNO in HEN solution overnight at 4 °C in the dark. Excess GSNO was removed using Bio-Spin Tris columns (Bio-Rad).

Gel-based RNase A reoxidation analysis was performed as described previously[45]. 2 μM PDI9, 2 μM ERO1 or SNO-ERO1, 100 μM FAD and 8 μM denatured and reduced RNase A were mixed and incubated in 50 μl reaction buffer (100 Tris-HAc, pH 8.0, 50 mM NaCl, and 1 mM EDTA) at 25 °C. At the indicated time points, the reaction was terminated by the addition of 5 × SDS loading buffer and 10 mM 4-acetamido-4'-maleimidylstilbene-2,2' disulfonic acid (AMS) to block the free thiols. The samples were analyzed by non-reducing SDS PAGE and stained with Fast Protein Stain (Biofuraw, UK).

### Detection of disulfide bonds by mass spectrometry
Disulfide bonds analysis for ERO1 was performed as described previously[84]. 20 μg recombinant ERO1 protein with or without GSNO treatment was directedly digested by trypsin, or trypsin/Glu-C on 10 kDa filters. Peptides were collected by centrifugation and dried in a centrifugal vacuum concentrator (Labconco, USA) for LC-MS/MS analysis.

After quantified by BCA kit, the total peptides were resuspended in 0.1% (v/v) formic acid solution, and then analyzed by an Orbitrap Fusion mass spectrometry (Thermo Fisher Scientific). The peptides were separated by Easy-nLC 1200 (Thermo Fisher Scientific), equipped with 150 mm in-house C18 analytical column. The peptide mixture was separated on the analytical column with the following conditions at 300 nl/min, 90 min run with a liquid gradient from 5–40% of solvent B (80% acetonitrile/0.1% FA). The parameters for a full MS survey scan were set as a resolution of 60,000 at 400 $m/z$ over the $m/z$ range of 300–1500, Automatic gain controls (AGC) target of 2E5, maximum ion injection time (IT) of 60 ms. The top 20 multiply charged parent ions were selected by data dependent MS/MS mode, fragmented by the high energy collision-induced dissociation (HCD). For the MS/MS detection, the resolution was set at 15,000, AGC target value was 1E5, and the maximum IT was 120 ms. Dynamic exclusion was enabled for 60 s. Data acquisition was controlled by Xcalibur (version 3.0, Thermo Fisher Scientific).

The raw files were submitted to P-link software (version 2.3) for disulfide bond identification and searched against the AtERO1 protein sequence. The identified disulfide bonds in ERO1 were manually inspected. The following parameters were used: oxidation of Met was set as variable modifications, and a maximum of three missed cleavage was allowed. The minimum peptide length was set at four amino acids. Mass tolerances were set up to 10 ppm for MS ions and 0.1 Da for MS/MS fragment ions. The false discovery rates of peptides were set at 1% FDR.

### Sequence alignment
The proteins used for sequence alignment were NP_177372.1 (*Arabidopsis thaliana*), XP_022550937.1 (*Brassica napus*), NP_001148525.1 (*Zea mays*), XP_002317004.3 (*Populus trichocarpa*), XP_015628190.1 (*Oryza sativa*), NP_001304609.1 (*Glycine max*), NP_055399.1 (*Homo sapiens*), NP_056589.1 (*Mus musculus*), NP_647865.2 (*Drosophila melanogaster*), and NP_013576.1 (*Saccharomyces cerevisiae*). The alignment was performed with MUSCLE (https://www.ebi.ac.uk/Tools/msa/muscle).

### Reporting summary
Further information on research design is available in the Nature Portfolio Reporting Summary linked to this article.

## Data availability
The mass spectrometry proteomics data have been deposited to the ProteomeXchange Consortium via the PRIDE partner repository with the dataset identifier PXD037504. Source data are provided with this paper.

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

## Acknowledgements

This work was supported by the National Key Research and Development Program of China, Grant 2021YFA1300402 to P.W. and National Institutes of Health, Grant 3RF1AG064250 to W.A.T. We are grateful to Drs. Dongping Lu of Center for Agricultural Resources Research, University of the Chinese Academy of Sciences for providing the mutants and constructions of *ERO1*, and Andreas Meyer of University of Bonn for helpful discussion.

## Author contributions

G.Q., W.A.T., and P.W. designed the research. G.Q., M.Q., and B.J. performed the experimental studies. G.Q., W.W., C.-P.S., and P.W. carried out the analysis. G.Q., Z.L., W.A.T., and P.W. wrote the manuscript.

## Competing interests

The authors declare no competing interests.
