## [Peer Review File · Nature Communications]

FAT-switch-based quantitative S-nitrosoproteomics reveals a key role of GSNOR1 in regulating ER functionsReviewer #1 (Remarks to the Author):

The authors developed a new method for S-nitrosylation peptide enrichment and detection in Arabidopsis and revealed a novel role of GSNOR1 or protein S-nitrosylation in regulating ER function. As not familiar with the approaches, so I did not criticize this part in detail. However, the conclusion on the key role of GSNOR1 in ER function regulation is not convincing.

Major points:

1. Several well-known S-nitrosylated proteins such as NPR1, RBOHD were not present in this S-nitrosoproteomics results of hot5/gsnor1/par mutants. This result challenges the sensitivity of FAT-switch-method for S-nitrosoproteomics in plants.
2. Based on the nitrosoproteomics, the authors found many ER-related proteins changed in gsnor1 mutant, why only the ERO1 was selected to identify and construct the relationship between ER stress and S-nitrosylation. Moreover, it is not clear about the regulation role of ERO1 in the tolerance phenotype of gsnor1 mutant in response to ER stress? Was S-nitrosylation levels of ERO1 detected in hot5-4 mutant and WT under Tm treatment?
3. Only one line of gsnor1/hot5/par2 mutant was selected to test the tolerance phenotype of ER stress under TM treatment. The result is not convincing. This experiment should be done with different hot5/gsnor1/par mutant lines or in the complementary lines. Different hot5/gsnor1/par mutant lines and the complementary lines have been already established in previous studies.
4. There have been several reports on the detection of changes in the S-nitrosylation level of gsnor mutants in various species. Only the method has been improved, raising concern about whether the application of the method is innovative.
5. Authors claimed they revealed an unreported function of protein S-nitrosylation in regulating ER functions. However, the regulation of protein S-nitrosylation of in ER stress has been reported in several previous studies (Nakato et al., 2015, Forrester et al., 2007), but it has not been mentioned in this study.

Minor points:

The information of the sentence from line 246 to line 247 is not correct. It has been reported that vacuolar H⁺-ATPase (AVP1) is not involving auxin transport or signaling.

TM and Tm should be identical throughout the manuscript.

Reviewer #2 (Remarks to the Author):

Qin et al, FAT-switch-based quantitative S-nitrosoproteomics reveals a key role of GSNOR in regulating ER functions

This manuscript presents a novel method for detection of protein S-nitrosylation in plants, a very newsworthy topic of NO research. Basically, the article is in excellent shape. Some comments:

The authors write in the Abstract that it would be challenging to quantitatively determine the S-nitrosylated targets in vivo. Exactly. That statement is of course absolutely correct. However, a correct quantification is probably only the technical part of the truth. Much more important might be the question regarding the biological meaning of any posttranslational modification (PTM). The authors do not distinguish between a gain-of-function and a loss-of-function effect of S-nitrosylation. A 5% S-nitrosylated share of the pool might be completely irrelevant if this PTM results in loss of activity. Maybe even a tenth of the activity is enough. On the other hand, if S-nitrosylation brings up any given activity, even a 1% share of the protein pool might be relevant. Please give these problem some room in the discussion.

The authors report on the global S-nitrosylation profiles in wild-type Arabidopsis. Among the S-nitrosylated sites, 408 sites in 360 protein groups were reported to play a role in redox homeostasis, system development, endoplasmic reticulum (ER) function and stress adaptation. The authors cite that about half of the S-nitrosylated peptides in Fares et al. (2011), Puyaubert et al. (2014) and in Lawrence et al. (2020) were included in the list of S-nitrosylation peptides identified by the FAT-method in their study (Supplementary Dataset S4). Please, where is the citation of the 2005 Lindermayr Plant Physiology paper? They were first to show a list of S-nitrosylated proteins in plants, with a method not so distinct from the one developed by Qin et al. In addition, apart from not citing this work (published 18 years earlier) it is crucial to compare the results. Which proteins are found by both methods? It is absolutely essential, that Qin et al show up with this comparison.

In addition to their interesting new method, Qin et al present a case study on S-nitrosylation of an oxidoreductase. Biochemical and genetic validation revealed that S-nitrosylation at Cys337 in ER OXIDOREDUCTASE 1 (ERO1) causes the rearrangement of disulfide, resulting in enhanced ERO1 activity. Consistently, the gsnor1 mutant shows improved tolerance to ER stress and is associated with the induced expression of unfolded protein response genes. Anyway, I'm not convinced these data need to be integrated in such a prominent way. I guess that about half of the results section is in this case study. Maybe the authors are aware of this problematic relation. However, they are no experts and if they are interested to push this part of the story they should cooperate with groups deeply digging in oxidoreductases or GSNOR.

Reviewer #3 (Remarks to the Author):

In this work, Qin et al developed a new method called FAT-switch for global and site-specific profiling of protein S-nitrosylation in plants. Compared to previous methods (e.g., Biotin switch), this newly developed method is much more sensitive. The authors deployed the FAT-switch method to identify more than 2,000 unique S-nitrosylated cysteine sites in 1,500 proteins in Arabidopsis seedlings, representing a several-fold improvement compared to previous studies in terms of the depth of S-nitrosylation profiling in plants. Furthermore, using the combination of FAT-switch and TMT, they made a quantitative comparison of S-nitrosyproteome in wild type and gsnor1, revealing wide-ranging regulation by this key redox regulator in plants. Taken together, from a redox proteomics perspective, this work is well done and should be a valuable resource for the plant community.

Point-to-point response to reviewer's comments

Reviewer #1 (Remarks to the Author):

The authors developed a new method for S-nitrosylation peptide enrichment and detection in Arabidopsis and revealed a novel role of GSNOR1 or protein S-nitrosylation in regulating ER function. As not familiar with the approaches, so I did not criticize this part in detail. However, the conclusion on the key role of GSNOR1 in ER function regulation is not convincing.

Major points:

(Reviewer #1 comment 1) 1. Several well-known S-nitrosylated proteins such as NPR1, RBOHD were not present in this S-nitrosproteomics results of *hot5/gsnor1/par* mutants. This result challenges the sensitivity of FAT-switch-method for S-nitrosproteomics in plants.

Response: We noticed the NPR1 and RBOHD were not present in our list of S-nitrosylation proteins by the FAT-switch method. Interestingly, they were not in any reported S-nitrosproteomics results (see Supplementary Dataset 4). LC-MS/MS-based proteomics occasionally miss some PTM targets identified by low-throughput studies, which might be because of extremely low abundance of a given modified peptide in the tested samples, or the modified peptides are not suitable for LC-MS/MS detection (too short or long peptide, inappropriate charges, etc.). For example, no NPR1 peptide was detected in the quantitative proteomics, suggesting its low abundance in the WT and *hot5/gsnor1* seedlings without any treatments. Missing RBOHD might be because its S-nitrosylation is induced by or only present upon immunity conditions (Yun et al., 2011), while we used the seedlings without any treatment. We added a sentence in the revision to clarify this point.

(Reviewer #1 comment 2) 2. Based on the nitrosproteomics, the authors found many ER-related proteins changed in *gsnor1* mutant, why only the ERO1 was selected to identify and construct the relationship between ER stress and S-nitrosylation.

Response: We worked on EROs because they have the conserved S-nitrosylation site (C337 in ERO1 and C336 in ERO2)(Fig. 4g), easy measurable enzyme activity *in vitro* (Fig. 3), and dysfunction of the protein resulted in apparent phenotype *in vivo* (Ugalde et al., 2022; Fan et al., 2019). Actually, our lab and colleagues currently work on validating other proteins selected by this criteria, like GSH1/PAD2. Though the data are also very promising, the work on ERO1 is the only one ready for publishing at this time. In addition, we feel the results on ERO1/2 are already strong enough to support the reliability of our proteomics result as a very detailed case study (Reviewer 2 already reminded us that the validation part is too much for a method and proteomics study, see **Reviewer #2 comment 3**). Thus, we would like to submit the works on other proteins separately after collecting all the necessary results.

(Reviewer #1 comment 3) Moreover, it is not clear about the regulation role of ERO1 in the tolerance phenotype of *gsnor1* mutant in response to ER stress? Was S-nitrosylation levels of ERO1 detected in *hot5-4* mutant and WT under Tm treatment?

Response: We appreciate the insightful comment from the reviewer and agree that the Tm-tolerant phenotype of *gsnor1* is intriguing. As presented in Table 1, our data suggest that besides ERO1, the dysfunction of GSNOR1 leads to the accumulation of S-nitrosylation in 16 other proteins involved in ER function, including proteins regulating protein glycosylation such as EBS1/UGGT, HAP6/RPN2, and AT2G39630. Therefore, multiple S-nitrosylation targets might comprehensively determine the Tm-tolerant phenotype of the *gsnor1* mutant. Consistent with this notion, we observed a significant increase in the expression of UPR genes in the *gsnor1* mutant compared to the WT, even without any treatment (Fig. 5f-h), indicating the activation of ER stress in the mutant, and may result in Tm sensitivity.

As we mentioned in our response to **Reviewer #1 comment 2**, we used ERO1/2 and ER stress as a case study to validate the S-nitrosylation proteins/sites identified by our FAT-switch method. The biochemistry data in Fig. 3 and 4 strongly support the C337 in ERO1 as a functional site crucial for its enzyme activity, indicating the reliability of the proteomics data.

Nevertheless, we attempted to generate the *gsnor1/ero1* double mutant by crossing *hot5-2* with *ero1-3*, but only obtained one *gsnor1/ero1* double mutant seedling from more than 150 F2 populations. Unfortunately, the *gsnor1/ero1* seedling showed severe infertility, suggesting that the knockout of ERO1 in the *hot5-2* mutant may cause more severe defects than *hot5-2* (see the attached figure below). Due to the very low fertility of the *hot5-2* mutant, it may take several months or even more than a year to generate stable transgenic lines with appropriate reduced expression of ERO1 by RNAi, or to generate the tagged-ERO1 transgene, in the *hot5-2* mutant background (as no commercial antibody for Arabidopsis ERO1/2 is available). Therefore, we are unable to fully address the reviewer's question in this manuscript. We would like to, as suggestion from **Reviewer #2 comment 3**, collaborate with other groups/experts to identify the major player responsible for the Tm-tolerant phenotype in the future.

a. The seedling and siliques of *hot5-2/ero1-3* and *hot5-2* mutants, b. Genotyping of the *hot5-2/ero1-3*.

(Reviewer #1 comment 4) 3. Only one line of *gsnor1/hot5/par2* mutant was selected to test the tolerance phenotype of ER stress under TM treatment. The result is not convincing. This experiment should be done with different *hot5/gsnor1/par* mutant lines or in the complementary lines. Different *hot5/gsnor1/par* mutant lines and the complementary lines have been already established in previous studies.

Response: We performed the parallel assays in the different alleles of *gsnor1/hot5/par2* and got similar results. We only presented the result of *hot5-4* (Ws background), a widely used allele of *gsnor1/hot5/par2*, in the original submission to consist of the Ws wild type and *hot5-4* samples used for proteomics. We thank the reviewer's suggestion and have now added the result of *hot5-2*, another allele of *gsnor1/hot5/par2* in Col-0 background, in the revised Supplementary Fig. 5.

(Reviewer #1 comment 5) 4. There have been several reports on the detection of changes in the S-nitrosylation level of *gsnor* mutants in various species. Only the method has been improved, raising concern about whether the application of the method is innovative.

Response: We appreciated the reviewer's comment. We chose *gsnor1*, a well-studied mutant accumulating high concentration of endogenous GSNO and S-nitrosylated proteins, to prove the sensitivity and efficiency of the FAT-switch method. As we emphasized in the manuscript, the FAT-switch method shows about 7.4- and 10-fold higher sensitivity and efficiency than the classic biotin-switch method, and it is comparable to the most efficient S-nitrosoproteomics method reported in animal cells by far (Mnatsakanyan et al., 2019). Additionally, this is the first quantitative nitrosoproteomics to compare the global S-nitrosylation in the wild type and a mutant and could be a powerful and applicable tool for measuring S-nitrosylation dynamics in different genetic varieties and/or different stimuli.

(Reviewer #1 comment 6) 5. Authors claimed they revealed an unreported function of protein S-nitrosylation in regulating ER functions. However, the regulation of protein S-nitrosylation of in ER stress has been reported in several previous studies (Nakato et al., 2015, Forrester et al., 2007), but it has not been mentioned in this study.

Response: We are sorry for the misleading. Actually, we mean this is the first study of S-nitrosylation targets in regulating ER functions IN PLANTS. In the third paragraph of DISCUSSION, we discussed the characterized S-nitrosylation in the animal studies: "In mammals, a few studies showed that NO-dependent protein modification is involved in ER functions. NO accumulation disturbs the Ca²⁺ homeostasis and induces ER stress responses, likely through tyrosine nitration of SERCA, and poly S-nitrosylation of cardiac calcium release channels. In brains manifesting sporadic Parkinson's or Alzheimer's disease, NO induces S-nitrosylation of PDI, which inhibits its enzymatic activity, leading to the accumulation of polyubiquitinated proteins and activation of the unfolded protein response. Recent studies showed that S-nitrosylation of a key UPR regulator, IRE1 α impairs ER function and prolongs ER stress in obesity. The discovery of novel S-nitrosylation targets in ER will reveal the molecular mechanism underlying ER stress responses and provide novel candidates to improve ER function and homeostasis upon stress or disease conditions in the future". The two previous studies the reviewer mentioned (Nakato et al., 2015, Forrester et al., 2007) still are works in animals. We thank the reviewer's reminder and correct the sentence in the revision.

Minor points:

(Reviewer #1 comment 8) The information of the sentence from line 246 to line 247 is not correct. It has been reported that vacuolar H⁺-ATPase (AVP1) is not involving auxin transport or signaling.

Response: We delete the vacuolar H⁺-ATPase (AVP1) from the sentence in the revision.

(Reviewer #1 comment 9) TM and Tm should be identical throughout the manuscript.

Response: We changed the TM to Tm throughout the manuscript.

Reviewer #2 (Remarks to the Author):

Qin et al, FAT-switch-based quantitative S-nitrosoproteomics reveals a key role of GSNOR in regulating ER functions
This manuscript presents a novel method for detection of protein S-nitrosylation in plants, a very newsworthy topic of NO research. Basically, the article is in excellent shape. Some comments:

(Reviewer #2 comment 1) The authors write in the Abstract that it would be challenging to quantitatively determine the S-nitrosylated targets in vivo. Exactly. That statement is of course absolutely correct. However, a correct quantification is probably only the technical part of the truth. Much more important might be the question regarding the biological meaning of any posttranslational modification (PTM). The authors do not distinguish between a gain-of-function and a loss-of-function effect of S-nitrosylation. A 5% S-nitrosylated share of the pool might be completely irrelevant if this PTM results in loss of activity. Maybe even a tenth of the activity is enough. On the other hand, if S-nitrosylation brings up any given activity, even a 1% share of the protein pool might be relevant. Please give these problem some room in the discussion.

Response: We agree with the reviewer on this point: discovering the biological effects of S-nitrosylation on the target proteins is equal to or more important than quantifying the PTM on a particular target protein. However, it has been well-documented that S-nitrosylation regulates activity, protein abundance, localization, turn-over, or protein-protein interaction of the target proteins. Currently, no high-throughput methods can measure all the biological changes at the proteome level. In addition, we want to emphasize that finding putative dominant targets from hundreds or even thousands of unrelative proteins is sometimes the most challenging task. Screening the mutation- or condition-responsive proteins by quantitative proteomics is a promising strategy for finding the candidates in a particular biological process. We modified the related sentences in the Abstract and Discussion in the revision according to this comment.

(Reviewer #2 comment 2) The authors report on the global S-nitrosylation profiles in wild-type Arabidopsis. Among the S-nitrosylated sites, 408 sites in 360 protein groups were reported to play a role in redox homeostasis, system development, endoplasmic reticulum (ER) function and stress adaption. The authors cite that about half of the S-nitrosylated peptides in Fares et al. (2011), Puyaubert et al. (2014) and in Lawrence et al. (2020) were included in the list of S-nitrosylation peptides identified by the FAT-method in their study (Supplementary Dataset S4). Please, where is the citation of the 2005 Lindermayr Plant Physiology paper? They were first to show a list of S-nitrosylated proteins in plants, with a method not so distinct from the one developed by Qin et al. In addition, apart from not citing this work (published 18 years earlier) it is crucial to compare the results. Which proteins are found by both methods? It is absolutely essential, that Qin et al show up with this comparison.

Response: We cited Lindermayr et al. (2005) paper in the Supplementary Table I, which summarized all nitrosoproteomics studies in plants by far. We omitted Lindermayr et al. (2005) in the original comparisons because the pioneer work 1) didn't contain the biotin-labeled peptide information, 2) used the **Accession Number** rather than **AGI ID** of proteins. Following the reviewer's suggestion, we manually checked the 105 proteins one by one and found that 41 (39%) out of 105 proteins are present in our result. It is a very impressive overlap between the pioneer study and our current result crossing 18 years. We added the comparison in the revised text and updated Supplemental Dataset 4.

(Reviewer #2 comment 3) In addition to their interesting new method, Qin et al present a case study on S-nitrosylation of an oxidoreductase. Biochemical and genetic validation revealed that S-nitrosylation at Cys337 in ER OXIDOREDUCTASE 1 (ERO1) causes the rearrangement of disulfide, resulting in enhanced ERO1 activity. Consistently, the gsnor1 mutant shows improved tolerance to ER stress and is associated with the induced expression of unfolded protein response genes. Anyway, I'm not convinced these data need to be integrated in such a prominent way. I guess that about half of the results section is in this case study. Maybe the authors are aware of this problematic relation. However, they are no experts and if they are interested to push this part of the story they should cooperate with groups deeply digging in oxidoreductases or GSNOR.

Response: As mentioned in our response to Reviewer #2 comment 1, we agree with the reviewer that studying the biological mean of PTM, is equal to or more important than quantifying or identifying the PTM on target proteins. That is the reason we pay much effort to biological validation parts. We verify whether the proteins in our list undergo S-nitrosylation *in vivo* (Figs 2b and 5a), and whether the S-nitrosylation sites we identified could function in plants (Figs. 3-5) using ERO1 as the case study. The promising results suggest the sensitivity and reliability of the FAT-switch method. Though I cannot say I am an expert in this field, we have worked on NO and S-nitrosylation for over 15

years. We published several papers on stress-regulated NO biosynthesis through NR phosphorylation (Wang et al., *Plant Cell*, 2010, 22, 2981-2998), S-nitrosylation of SnRK2s, the central components in ABA signaling (Wang et al., *PNAS*, 2015, 112, 613-618; *Plant Signal & Behav*, 2015, 10, e1031939). We know the method is the primary reason for imitating the study of S-nitrosylation in plants, and we are dedicated to developing this new proteomics method to overcome it.

We are happy the FAT-switch method worked fantastically and have already shared our method and data with some experts in this field, like Drs. Jianru Zuo of the Institute of Genetics and Developmental Biology and Andreas Meyer of the University of Bonn. We believe in the value of the FAT-switch method and the comprehensive list of S-nitrosylation targets, and are eager to share them with more scientists in the community.

Reviewer #1 (Remarks to the Author):

The authors have solved the problem as much as possible, I have no further comment now.

Reviewer #2 (Remarks to the Author):

OK! You made the right corrections.